# Shaping a Stabilized Video by Mitigating Unintended Changes for Concept-Augmented Video Editing

## Abstract

Text-driven video editing utilizing generative diffusion models has garnered significant attention due to their potential applications. However, existing approaches are constrained by the limited word embeddings provided in pre-training, which hinders nuanced editing targeting open concepts with specific attributes. Directly altering the keywords in target prompts often results in unintended disruptions to the attention mechanisms. To achieve more flexible editing easily, this work proposes an improved concept-augmented video editing approach that generates diverse and stable target videos flexibly by devising abstract conceptual pairs. Specifically, the framework involves concept-augmented textual inversion and a dual prior supervision mechanism. The former enables plug-and-play guidance of stable diffusion for video editing, effectively capturing target attributes for more stylized results. The dual prior supervision mechanism significantly enhances video stability and fidelity. Comprehensive evaluations demonstrate that our approach generates more stable and lifelike videos, outperforming state-of-the-art methods. The anonymous code is available at https://anonymous.4open.science/w/STIVE-PAGE-B4D4/.

## 1 Introduction

Text-driven video editing using generative diffusion models (Ho et al., 2020; Song et al., 2020; Rombach et al., 2021) has garnered significant attention due to its potential applications in various fields, including film production, art, and advertising (Ho et al., 2022; Hong et al., 2022; Blattmann et al., 2023).

Existing text-driven video editing methods based on diffusion models, such as Tune-A-Video (Wu et al., 2023), FateZero (Qi et al., 2023), Zeroscope (Sterling, 2023), and VideoComposer (Wang et al., 2024), have significantly improved the ability to edit objects, backgrounds, and styles in video scenes while maintaining overall scene consistency through the optimization of attention mechanisms and spatiotemporal continuity. These approaches have demonstrated notable success in video generation. However, they are often limited by the restricted word embeddings provided by CLIP (Radford et al., 2021) during the text-driven encoding process, which restricts their ability to perform diverse and nuanced edits on targets with specific attributes. Modifying words in the target prompt can disrupt the attention mechanisms, leading to inconsistencies in non-target areas before and after the editing process.

To address these limitations, recent methods such as (Bar-Tal et al., 2022; Lee et al., 2023; Chai et al., 2023) have explored the use of Neural Layered Atlases (NLA) (Kasten et al., 2021). These methods primarily focus on extracting layered atlases from video frames, editing these atlases using text-driven image-editing diffusion models (Rombach et al., 2021; Zhang et al., 2023), and then synthesizing the final edited video through post-processing. While this approach is effective at preserving non-edited background areas, it exhibits poor performance in maintaining spatio-temporal continuity. Moreover, the processing of individual image frames makes the generation of neural atlases extremely time-consuming.

To achieve more diverse editing results easily, one feasible approach is to draw inspiration from the Textual Inversion (Gal et al., 2022) method used in image generation by incorporating external

concept word embeddings. In text-to-image generation, external word embeddings (referred to as "concepts") are optimized within the CLIP text encoder (Radford et al., 2021) while freezing the diffusion model's parameters. This allows the model to address the need for user-provided custom images to guide image editing. However, using a pre-trained diffusion model for self-supervised textual inversion presents limitations in the expressive power of these external embeddings, which are constrained by the size of the latent space and the number of training iterations, often leading to under-fitting and restricted expressive capacity.

In this paper, we propose an improved concept-augmented video editing method. This approach flexibly generates diverse and stable target videos by defining abstract conceptual pairs (concept prompt and concept video) that describe the target scene. Specifically, we propose the **Concept-Augmented Textual Inversion** method, which reliably and accurately captures the target attributes in the user's custom video. In addition, we also introduce a **Dual Prior Supervision** mechanism that stabilizes the generated video by crossing the attention between the source and target, preventing attention dispersion caused by modifications to the target prompt. This mechanism effectively improves the consistency of non-target areas before and after video editing, while also enriching the fidelity of the concepts in the edited results. Our key contributions are as follows:

- We orchestrate a framework that allows users to extract concepts from custom videos to efficiently generate diverse edited videos through concept templates.
- We propose a concept-augmented textual inversion method, which efficiently and stably extracts detailed attributes of the target in the user's custom concept video and supports plug-and-play guiding of stable diffusion for video editing.
- We present a dual prior supervision mechanism, which effectively improves the consistency and stability of video editing results.

## 2 RELATED WORK

**Text-Driven Video Synthesis.** A series of works based on diffusion models (Ho et al., 2020; Song et al., 2020; Rombach et al., 2021) has made significant progress in text-driven image generation. Subsequent efforts (Esser et al., 2023; Wang et al., 2023; Blattmann et al., 2023) further achieve text-driven video generation by extending existing image generation diffusion models. These works commonly inherit the spatial parameters of UNet and fine-tune the newly added temporal modules with large-scale video-text pair datasets to improve inter-frame stability for video synthesis. These works laid a good foundation for video editing with textual descriptions.

**Text-Driven Video Editing.** Current approaches for text-driven video editing mainly fall into three categories: fine-tuning video generation models (Zhao et al., 2023; Wang et al., 2024), fine-tuning image generation models extended with temporal modules (Wu et al., 2023; Qi et al., 2023), and combining NLA (Kasten et al., 2021) with pre-trained image generation models (Bar-Tal et al., 2022; Lee et al., 2023; Chai et al., 2023). MotionDirector (Zhao et al., 2023) improves the ability of editing camera and object motions by adding LoRA (Hu et al., 2022) to the attention modules of the pre-trained Zeroscope (Sterling, 2023), strengthening the connection between texts and motions in edited videos. VideoComposer (Wang et al., 2024) enhances inter-frame consistency by introducing a condition fusion module with spatial and temporal conditions such as motion vectors, depth maps, and sketches. Recent advances have demonstrated various innovative approaches in these categories. For example, (Ku et al., 2024) employs a pretrained model for diverse video editing tasks, while GenVideo (Singer et al., 2025) utilizes a target-image-aware approach with novel InvEdit masks to overcome text-prompt limitations. Besides, (Singer et al., 2025) introduces the EVE model by distilling pretrained diffusion models. (Bar-Tal et al., 2022; Lee et al., 2023; Chai et al., 2023) extract layered neural atlases from video frames to edit atlases which are further processed to synthesize videos; however, generating a neural atlas demands considerable computational time.

Recently, Tune-A-Video (Wu et al., 2023) achieves one-shot video editing with improved inter-frame coherency by updating self-attention with sparse causal attention. FateZero (Qi et al., 2023) further proposes self-attention blending and incorporates attention control (Hertz et al., 2023) to enhance the ability of editing objects, background, and styles while maintaining scene consistency. For temporal consistency specifically, VidToMe merges self-attention tokens across frames, while (Geyer et al., 2023) leverages inter-frame correspondences to propagate features. In spatial editing,

approaches like (Ceylan et al., 2023; Cohen et al., 2024; Liu et al., 2024) improve results using spatial or temporal attention features and diffusion models. For editing targets with specific attributes, it becomes necessary to introduce external word embeddings. Our method supports the incorporation of external concept word embeddings. Furthermore, inspired by Tune-A-Video (Wu et al., 2023) and FateZero (Qi et al., 2023), we introduce a dual prior supervision mechanism between video frame latents and word embeddings to enhance scene consistency before and after video editing based on the prompt-to-prompt attention control method. Compared to existing approaches, our method focuses on attention supervision and control mechanisms and operates on a one-shot video editing paradigm. It also improves temporal consistency through extended temporal module parameters and enables the flexible integration of external concept objects, while CLIP-based (Radford et al., 2021) methods above are constrained by finite word embeddings.

**Textual Inversion.** (Gal et al., 2022) proposes a textual inversion method that optimizes newly added concept word embeddings in the CLIP (Radford et al., 2021) text encoder, supervised by the latent variable distribution of specific images in the diffusion model. However, using a pre-trained diffusion model for self-supervised text inversion may lead to under-fitting for some specific images due to the finite latent variable space. Although it's feasible to optimize concept word embeddings with a smaller learning rate simultaneously, or to train the diffusion model with frozen concept embeddings in the next stage, this process faces issues of easy over-fitting and high storage costs. Our method, building upon textual inversion (Gal et al., 2022), attempts to add LoRA (Hu et al., 2022) modules to the diffusion model, optimizing them simultaneously with concept words at a smaller learning rate, to enhance the text editing capabilities of concept words.

**Cross Attention Control and Supervision.** Prompt-to-Prompt (Hertz et al., 2023) proposes three attention control methods for stable text-driven image editing based on diffusion models: word swap, refinement, and reweighting. By applying the cross-attention probability map recorded from the original text and original image latent variables to the denoising process of edited text and original image latent variables, it has achieved significant success in stable text-driven image editing (Avrahami et al., 2022; 2023). Additionally, (Qi et al., 2023) proposed self-attention blend effectively transfers the stability of text-driven image editing to the video editing domain. Our method, building upon this foundation, introduces external concept words to support editing with higher degrees of freedom. However, when performing text-driven editing, whether using existing word embeddings or external concept word embeddings as editing words, there exists a problem of attention dispersion. This means that editing words have non-negligible effects on latent variables other than the editing target. Inspired by the work of (Yang & Tang, 2022), we introduce an attention supervision mechanism to address the issue of dispersed attention in editing words.

## 3 METHOD

### 3.1 PRELIMINARIES

**Textual Inversion.** Textual inversion (Gal et al., 2022) learns new embeddings that represent user-provided visual concepts within the textual embedding space. These learned embeddings are then associated with pseudo-words that can be incorporated into novel sentences to achieve text-to-vision editing. The learning process of textual inversion relies on a latent variable diffusion model, which typically consists of an autoencoder and a noise prediction network. For an image $x$, the autoencoder is pretrained such that the encoder $\mathcal{E}$ maps the image to a latent variable $z = \mathcal{E}(x)$, and the decoder $\mathcal{D}$ reconstructs the original image from the latent variable $x \approx \mathcal{D}(z)$. Particularly, textual inversion leverages a CLIP (Radford et al., 2021) text encoder $c_\theta$ with additional concept words to encode conditional text input $y$. The optimization objective is defined as:

$$\mathcal{L}_{noise} = \mathbb{E}_{z \sim \mathcal{E}(x), y, \epsilon \sim \mathcal{N}(0,1), t}[\|\epsilon - \epsilon_\theta(z_t, t, c_\theta(y))\|_2^2], \tag{1}$$

where $z_t$ is the noised latent at time step $t$, $\epsilon$ is the noise, and $\epsilon_\theta$ is the noise prediction network.

**Low-Rank Adaption.** (Hu et al., 2022) proposes an efficient fine-tuning scheme based on matrix low-rank decomposition. For the pre-trained weight $W_0 \in \mathbb{R}^{d \times k}$ in the original model, it applies low-rank decomposition to update the weight as $W = W_0 + \Delta W$, where $\Delta W = BA$, $B \in \mathbb{R}^{d \times r}$, $A \in \mathbb{R}^{r \times k}$, and $r \ll min(d, k)$. During the fine-tuning process, the pre-trained weight $W_0$ is frozen, while $A$ and $B$ are trainable parameters. For the forward computation of the original weight

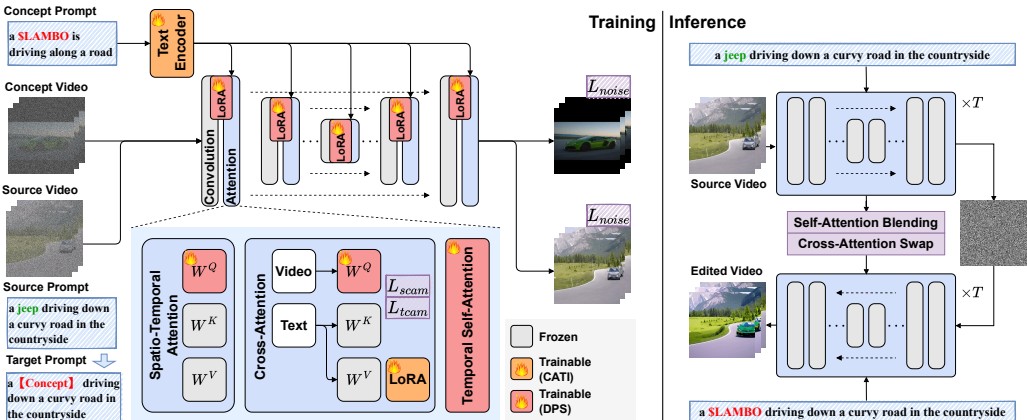

Figure 1: **Overview of our training and inference pipelines.** During the training stage, we first adapt the diffusion model to new visual concepts using our introduced Concept-Augmented Textual Inversion (CATI), and then we tune the temporally extended diffusion model with our proposed Dual Prior Supervision (DPS) mechanism to prevent unintended changes in edited videos. During the inference stage, we blend self-attention matrices to retain semantic layout (Self-Attention Blending) and swap cross-attention matrices to achieve text-driven video editing (Cross-Attention Swap).

$h = W_0 x$, the updated forward computation becomes:

$$LoRA(h) = W_0 x + \Delta W x. \tag{2}$$

**Video Diffusion Models with Temporal Extensions.** Tune-A-Video (Wu et al., 2023) introduces Spatio-Temporal Attention (ST-Attn) to replace the original Self-Attention (Vaswani, 2017) in the 2D UNet. When calculating the keys $K$ and values $V$, ST-Attn concatenates latent variables of the first and former frames of the video, leading to the attention result where the current frame attends to both the first and former frames. The specific operations for replacing $K, V$ in Self-Attention are as follows:

$$K = W^K[z_{v_1}; z_{v_{i-1}}], V = W^V[z_{v_1}; z_{v_{i-1}}], \tag{3}$$

where $W^K$ and $W^V$ are projection matrices for key and value respectively, $z_{v_i}$ denotes the latent variable of the $i$-th frame of the video to the current attention layer, and $[\cdot]$ denotes concatenation.

### 3.2 STABILIZED TEXT-DRIVEN VIDEO EDITING

Our training and inference pipelines are visualized in Fig. 1. We adopt a UNet that inherits the pre-trained 2D UNet parameters from Stable Diffusion (Rombach et al., 2021) as the noise prediction network. The original 2D UNet consists of a series of spatial convolution layers and cross-attention layers. To adapt it for 3D video inputs, we replace the original spatial self-attention layers with spatio-temporal attention as explained in Eq. (3). Following FateZero (Qi et al., 2023), we also incorporate LoRA-structured temporal convolution layers after the spatial convolution layers, and temporal self-attention layers with zero-initialized linear output layers after the cross-attention layers. The outputs of these newly added modules are residually connected to the outputs of the original modules.

Our approach for stabilized text-driven video editing has two learning phases. In the first phase, we introduce Concept-Augmented Textual Inversion (CATI) to adapt the diffusion model to new visual concepts. In the second phase, we tune partial parameters of the temporally extended diffusion model to suppress unintended changes in edited videos by calibrating cross-attention results.

**Concept-Augmented Textual Inversion.** Textual inversion (Gal et al., 2022) learns to represent a specific set of user-provided images with pseudo-words in the latent space, offering an intuitive way for natural language-guided image editing. We incorporate this technique into our framework to facilitate video editing. However, due to the self-supervised nature within the limited latent space of the pre-trained diffusion model, the vanilla textual inversion often results in varied performance in terms of quality and efficiency for different image sets, requiring meticulous adjustments for learning rates and iteration counts.

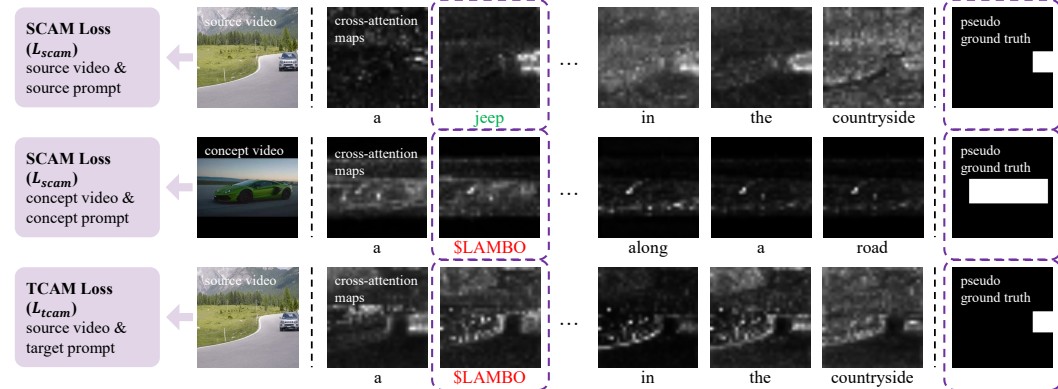

Figure 2: **Visualization of the dual prior supervision mechanism.** Each row displays a video frame, a set of cross-attention maps between this video frame and prompt words, and a pseudo ground truth mask. The *scam* loss and *tcam* loss are computed between relevant words and pseudo masks to prevent unintended changes for stabilized video editing.

To alleviate this issue, we draw inspiration from existing parameter-efficient fine-tuning techniques and propose adding LoRA modules (Hu et al., 2022) to the value projection parameters in the cross-attention layers of the UNet. Consequently, the values $V$ are updated to $LoRA(V)$ according to Eq. (2). The rationale behind our approach is that we aim to enhance the expressiveness of the pre-trained diffusion model by slightly adjusting its capacity to accommodate new visual concepts while preserving its original generation capability. Besides, inserting LoRA modules not only augments textual inversion with low storage overhead but also maintains a plug-and-play characteristic during inference. We train the concept-word embeddings of textual inversion and the weight parameters of LoRA modules in an end-to-end manner (see orange blocks in Fig. 1), where the learning rate for LoRA parameters is relatively smaller than that for concept-word embeddings to avoid over-fitting. Denote the noise prediction network with LoRA modules loaded on value projection parameters as $\epsilon_{\theta_L}$, the optimization objective of concept-augmented textual inversion is then updated from Eq. (1) to the following:

$$\mathcal{L}_{noise} = \mathbb{E}_{z \sim \mathcal{E}(x), y, \epsilon \sim \mathcal{N}(0,1), t}[\|\epsilon - \epsilon_{\theta_L}(z_t, t, c_\theta(y))\|_2^2]. \tag{4}$$

**Model Tuning with Dual Prior Supervision.** After learning concept-augmented textual inversion, we adapt and tune the video diffusion model for text-driven video editing in line with the paradigm of few-shot learning. Specifically, we learn the LoRA-structured temporal convolution layers, the query projection weights within spatio-temporal attention layers and cross-attention layers, and the temporal self-attention layers (see red blocks in Fig. 1). These parameters are selected for updates during training due to their strong relevance to the temporal modeling of 3D videos. To attain more stable and higher quality editing results, we tried directly integrating existing attention control techniques (Hertz et al., 2023) in an early attempt; however, we found that when applying text-driven video editing types such as word swap, the dispersion phenomenon of cross-attention between text embeddings and video latents leads to reduced stability in editing results. To address this challenge, we propose a dual prior supervision mechanism, which includes a source cross-attention mask (*scam*) loss and a target cross-attention mask (*tcam*) loss.

The *scam* loss is designed to reduce the attention influence of the words to be replaced in the source prompt on irrelevant frame areas (see the first row in Fig. 2). It is also applied to modulate attention between concept words and concept videos (see the second row in Fig. 2). Specifically, for $K$ cross-attention layers in the UNet, we record cross-attention matrices $\mathbf{M}_s$ between the words and the video frame latents in each cross-attention layer. To obtain ground truth for optimization, we use an off-the-shelf object detection network OWL-ViT (Minderer et al., 2022) to localize objects in video frames and generate corresponding binary pseudo-labels $\mathbf{M}_s^{\text{gt}}$. We further apply max pooling to generate $K$ pseudo-labels, each with a designated resolution $P_k$. The loss is then calculated as the mean absolute loss on irrelevant areas:

$$\mathcal{L}_{scam} = \frac{1}{K} \sum_{k=1}^{K} \sum_{i=1}^{P_k} \left[ \|\mathbf{M}_{s,k,i}^{\text{gt}} - \mathbf{M}_{s,k,i}\| \cdot (1 - \mathbf{M}_{s,k,i}^{\text{gt}}) \right]. \tag{5}$$

The *tcam* loss is introduced to diminish the attention influence of the target words in the edited prompt to further promote the consistency of irrelevant areas before and after video editing (see the third row in Fig. 2). Similar to the *scam* loss, we obtain cross-attention matrices $\mathbf{M}_t$ and pseudo-labels $\mathbf{M}_t^{\text{gt}}$ between the target words in the edited prompt and the video frame latents. The loss is computed as:

$$\mathcal{L}_{tcam} = \frac{1}{K} \sum_{k=1}^{K} \sum_{i=1}^{P_k} \left[ \|\mathbf{M}_{t,k,i}^{\text{gt}} - \mathbf{M}_{t,k,i}\| \cdot (1 - \mathbf{M}_{t,k,i}^{\text{gt}}) \right]. \tag{6}$$

Let the trainable parameters during the model tuning phase be denoted as $\epsilon_{\theta_T}$. The noise prediction loss $\mathcal{L}_{noise}$ is then obtained by substituting $\epsilon_\theta$ in Eq. (1) with $\epsilon_{\theta_T}$. Given $\alpha$ and $\beta$ as the weighting coefficients for our proposed *scam* loss and *tcam* loss, respectively, the total loss for model tuning with dual prior supervision is formulated as:

$$\mathcal{L} = \mathcal{L}_{noise} + \alpha \mathcal{L}_{scam} + \beta \mathcal{L}_{tcam}. \tag{7}$$

**Inference.** As shown in Fig. 1, the inference pipeline involves an inversion stage using the source text prompt, and an editing stage using the modified text prompt. We cache the intermediate self-attention matrices and cross-attention matrices at each time step during inversion. These matrices are then leveraged to manipulate attention during editing. Specifically, we blend self-attention matrices to retain the semantic layout following FateZero Qi et al. (2023) (Self-Attention Blending), and swap cross-attention matrices between the changed words and video latents similar to Prompt-to-Prompt Hertz et al. (2023) (Cross-Attention Swap).

## 4 EXPERIMENTS

### 4.1 SETTINGS AND DATASETS

Our experiments are conducted on a machine equipped with an NVIDIA GeForce RTX 4090. During the concept augmented textual inversion stage, we set the learning rate for CLIP (Radford et al., 2021) word embeddings to $1 \times 10^{-3}$, and the learning rate for LoRA modules inserted into the UNet to $1 \times 10^{-5}$, with the number of training steps set to 5000. Additionally, we randomly sample frame numbers within the range $[4, 8]$ from the concept video during training, to prevent the inversion process from over-fitting to a fixed frame number. For the video diffusion model fine-tuning stage, we empirically set $\alpha = 0.1$ and $\beta = 0.1$ in Eq. (7). The training steps above all use AdamW (Loshchilov & Hutter, 2017) optimizer. In the inference stage of video editing, the guidance scale is set to 12.5, the number of DDIM Inversion steps is $T = 50$, and the self-attention blending and cross-attention swap steps are within the interval $[0, 0.7T]$. To evaluate our proposed method, we used a portion of the DAVIS (Caelles et al., 2019) dataset and clip videos from the internet to construct video editing pairs, either with or without concept videos.

### 4.2 METRICS

**Frame Consistency.** To compare the coherence of the video frames $\mathbb{F}$, we refer to the metric used in (Wu et al., 2023; Hessel et al., 2021), which calculates the average cosine distance $d$ between features $(\boldsymbol{v}_i, \boldsymbol{v}_j)$ of each two different frames $(\boldsymbol{f}_i, \boldsymbol{f}_j)$ encoded by the CLIP visual encoder (Radford et al., 2021), as Eq. (8). Here, $\boldsymbol{f}_i, \boldsymbol{f}_j \in \mathbb{F}$, $\boldsymbol{f}_i \neq \boldsymbol{f}_j$, and $\mathbb{D}$ denotes the set of the vector pairs $(\boldsymbol{v}_i, \boldsymbol{v}_j)$.

$$d = \frac{1}{|\mathbb{D}|} \sum_{(\boldsymbol{v}_i, \boldsymbol{v}_j) \in \mathbb{D}} \frac{\boldsymbol{v}_i \cdot \boldsymbol{v}_j}{\|\boldsymbol{v}_i\| \|\boldsymbol{v}_j\|}. \tag{8}$$

**Masked Peek-Signal-Noise Ratio.** To compare the stability of the video non-target areas before and after target editing, we design a Masked Peak Signal-to-Noise Ratio (**M-PSNR**) metric. We use the OWL-ViT (Minderer et al., 2022) open-vocabulary object detection model with text pseudo-labels to estimate the bounding box mask $M$ of the edited target. We then compare the average peek-signal-noise ratio of the original video frames and the edited video frames after applying this mask. The calculation formula for the specific function $f$ for the Mean Squared Error (MSE) used

as input is as follows, where $\boldsymbol{M} \in \mathbb{R}^{H \times W}$, $\boldsymbol{I}^s \in \mathbb{R}^{H \times W \times C}$, and $\boldsymbol{I}^e \in \mathbb{R}^{H \times W \times C}$ refer to the mask value, the frame pixel value of video before and after editing, respectively.

$$f(\boldsymbol{I}^s, \boldsymbol{I}^e, \boldsymbol{M}) = \frac{1}{C} \frac{\sum_{k \in C} \sum_{i \in H} \sum_{j \in W} (I^s_{i,j,k} - I^e_{i,j,k})^2 (1 - M_{i,j})}{\sum_{i \in H} \sum_{j \in W} (1 - M_{i,j})}. \tag{9}$$

**Concept Consistency.** To evaluate the correlation between the video editing results guided by the concept video and the concept video itself, while minimizing non-target areas interference, we employ a multi-step approach. First, we utilize a pre-trained OWL-ViT (Minderer et al., 2022) model in conjunction with pseudo-label prediction to generate object masks for both videos. Using these masks, we then extract pixel segments of the target objects from both the edited video and the concept video. Finally, we leverage the CLIP model to predict visual encoding vectors for these extracted segments and calculate the average cosine similarity between them.

### 4.3 COMPARISONS WITH EXISTING METHODS

**Quantitative Evaluation.** As illustrated in Tab. 1, we quantitatively assess text-driven video editing results in three aspects. Compared with existing methods that extend and fine-tune the Stable Diffusion model, including Tune-A-Video (Wu et al., 2023), FateZero (Qi et al., 2023), RAVE (Kara et al., 2024), and MotionDirector (Zhao et al., 2023), our approach demonstrates superior inter-frame coherence in terms of the Frame Consistency Metric. To evaluate the consistency of unrelated areas before and after video editing, we employ M-PSNR as a reference metric, and our method achieves the highest score by a large margin. Concretely, our method outperforms MotionDirector (Zhao et al., 2023) by a noticeable 6.98 M-PSNR in editing with concept video. This is attributed to our proposed prior supervision mechanism, which effectively reduces the editing noise in non-target areas for both source and concept videos. Furthermore, to evaluate the target fidelity in concept and edited videos, we utilize Concept Consistency as a reference metric, and our method demonstrates greater fidelity compared to others.

**Qualitative Evaluation.** The visual comparison results of video editing with and without concept video guidance are shown in Fig. 3. As can be seen, our method can maintain content consistency in non-target areas before and after video editing with and without concept videos. Particularly, when using concept videos, our method can effectively introduce the visual concept from the concept video into the edited video. For example, in Fig. 3 (Setting I), our method successfully replaces 'man' with the concept '$OPTIMUS', while others either fail to preserve the background or cannot transfer the integral target shape.

On the other hand, other approaches commonly face instability in non-target areas of their edited videos. Tune-A-Video (Wu et al., 2023) encounters the issue of dispersed cross-attention between word embeddings and video latents as it fine-tunes the model using only one video-text pair. While FateZero (Qi et al., 2023) and RAVE (Kara et al., 2024) mitigate this issue by manipulating cross-attention matrices or shuffling noise in a zero-shot manner, these methods directly use concepts to drive video editing, which results in compromised non-target area consistency and degraded concept fidelity. MotionDirector (Zhao et al., 2023) naturally supports extracting targets from concept videos via its trainable spatial path; however, the coupled spatial and temporal paths struggle to provide stable guidance, leading to noticeable inconsistencies in non-target areas. In contrast, our proposed concept-augmented textual inversion and dual prior supervision can effectively maintain content consistency in non-target areas before and after video editing while accurately capturing specific attributes of user-provided concepts.

| Method | Editing w/ Concept Video | | | Editing w/o Concept Video | |
| --- | --- | --- | --- | --- | --- |
| | M-PSNR ↑ | Concept Cons. ↑ | Frame Cons. ↑ | M-PSNR ↑ | Frame Cons. ↑ |
| Tune-A-Video (Wu et al., 2023) | 14.70 | 0.6982 | 0.9399 | 15.72 | 0.9397 |
| FateZero (Qi et al., 2023) | 17.08 | 0.6822 | 0.9413 | 19.42 | 0.9246 |
| MotionDirector (Zhao et al., 2023) | 12.73 | 0.7222 | 0.9452 | 16.86 | 0.9403 |
| RAVE (Kara et al., 2024) | 17.39 | 0.6990 | 0.9379 | 16.20 | 0.9306 |
| Ours | **19.71** | **0.7642** | **0.9472** | **22.10** | **0.9405** |

Table 1: **Quantitative comparison of video editing results.**

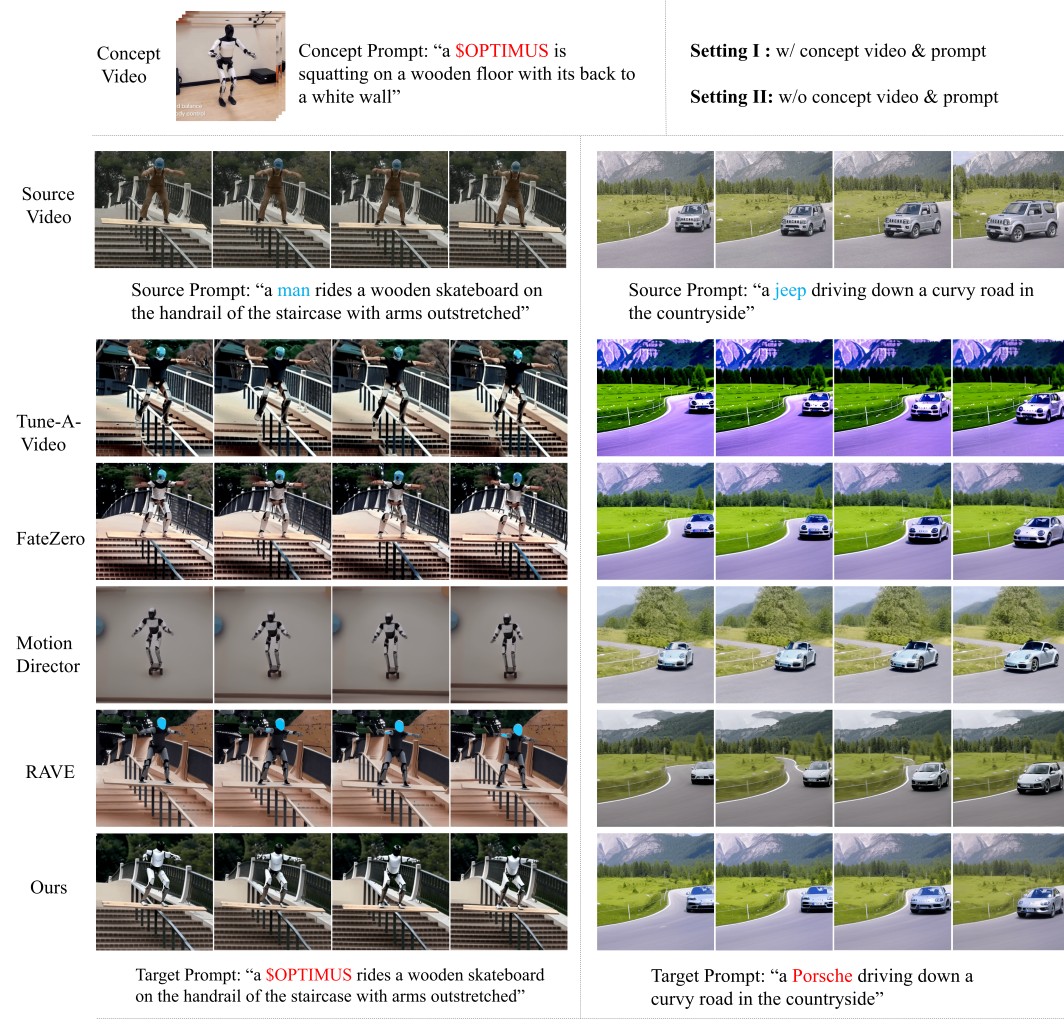

Figure 3: **Video generation with (Setting I) and without (Setting II) concept pairs.** The top row of the figure contains the concept video with its prompt in the left part, and comparison settings in the right part. The second row is the source video frames to be edited and its prompt. The rows below show the editing results of the source video using the editing prompt, for (Wu et al., 2023), (Qi et al., 2023), (Zhao et al., 2023), (Kara et al., 2024) and our method, respectively, in which words with '$' ahead mean concept words, and the same applies to subsequent results.

## 4.4 Ablation Study

**Concept Augmentation Alleviates Under-Fitting of Textual Inversion.** In this work, we draw on the idea of Textual Inversion (TI) from text-to-image generation and apply it to text-driven video editing to address the embedding of external concept words. However, simply applying TI may lead to under-fitting, resulting in a lack of realism. For instance, in the results shown in Fig. 4(a) and Fig. 4(c), where the keywords 'jeep' are altered to '$LAMBO' and '$CYBERTRUCK', although some attributes (e.g., shape) of the target concepts are partially retained, the results appear to "drift" due to insufficient inductive bias. In contrast, the concept-augmented textual inversion can effectively capture the color, shape, and other attributes, as demonstrated in Fig. 4(b) and Fig. 4(d). The concept augmentation provides more detailed features to the target, significantly improving the fidelity of details in the inversion results.

**Dual Prior Supervision Improves Stability and Fidelity.** In this work, we propose a Dual Prior Supervision strategy, which consists of two main components (See Sec. 3.2): *scam* loss and *tcam* loss. Both components play crucial roles in maintaining the stability of the target generation. By

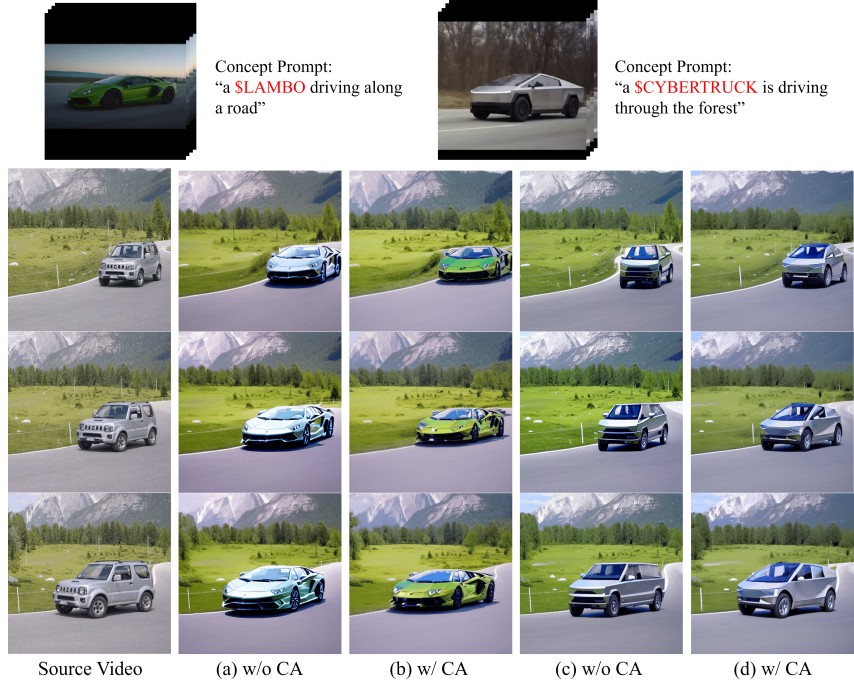

Figure 4: **Comparison of whether to use Concept Augmentation (CA) for textual inversion.** Compared the text inversion results without and with concept augmentation for pairs (a), (b): 'jeep' → '$LAMBO'; and (c), (d): 'jeep' → '$CYBERTRUCK', respectively, from the same source prompt "a jeep driving down a curvy road in the countryside".

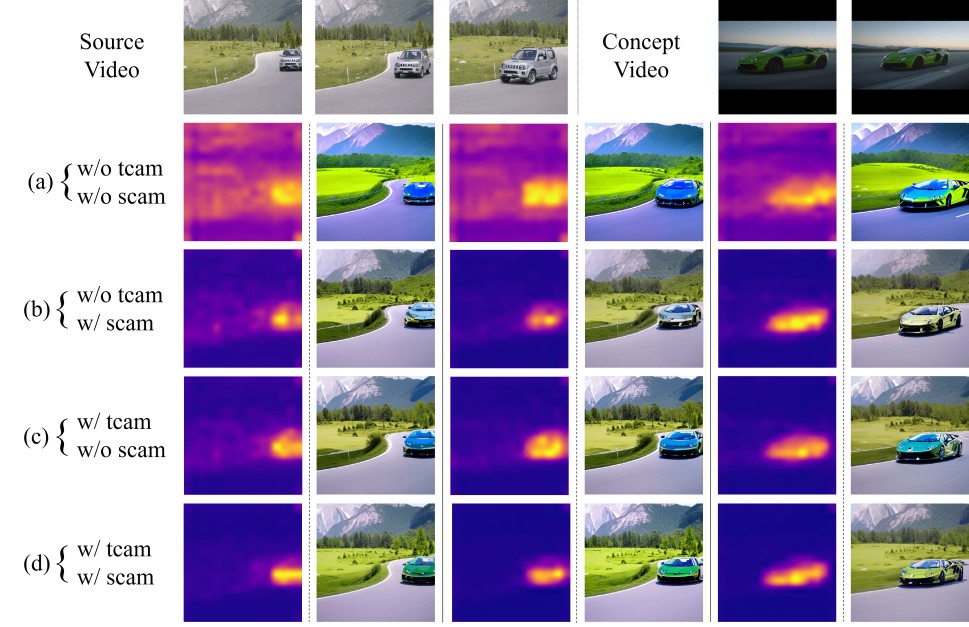

Figure 5: **The impact of dual prior supervision.** From the first to the last row, using the editing example in Fig. 1, we compare the average cross-attention maps and the editing results with and without the supervision mechanism of *scam* and *tcam*. Each case contains three pairs, and each pair consists of an average cross-attention map on the left and an edited frame on the right. The full comparisons are put in Fig. 11 and Fig. 12.

comparing the attention regions in Fig. 5 (a) (w/o *tcam*, w/o *scam*), Fig. 5 (b) (w/o *tcam*, w/ *scam*), and Fig. 5 (c) (w/ *tcam*, w/o *scam*), we can conclude that both *scam* and *tcam* (Fig. 5 (d)) significantly reduce background disturbances and improve stability. However, the generated video results reveal

that using either component alone cannot effectively reproduce the target object's attributes, such as the car's color. The proposed **Dual Prior Supervision**, which combines both components, not only enhances the stability of the background in the target results, but also captures the target object's attributes more accurately, thereby improving the fidelity of the edited concept target.

**Tuning w/ Concept Video Produces Stylized Results.** Recall that we construct the target videos in this work by templating the concept pairs to make the editing process more flexible. To explore the impact of the concept video in **Setting I** (Fig. 3), we conduct a simple experiment as shown in Fig. 6. As shown in Fig. 6(a) and Fig. 6(b), tuning models with both concept video and concept prompt produce more stylized videos. The possible explanation for this is that the introduction of the concept video alleviates the over-fitting issue. Note that tuning with only the concept video (without the concept prompt) is not viable here, as we cannot analyze the intent without any textual guidance.

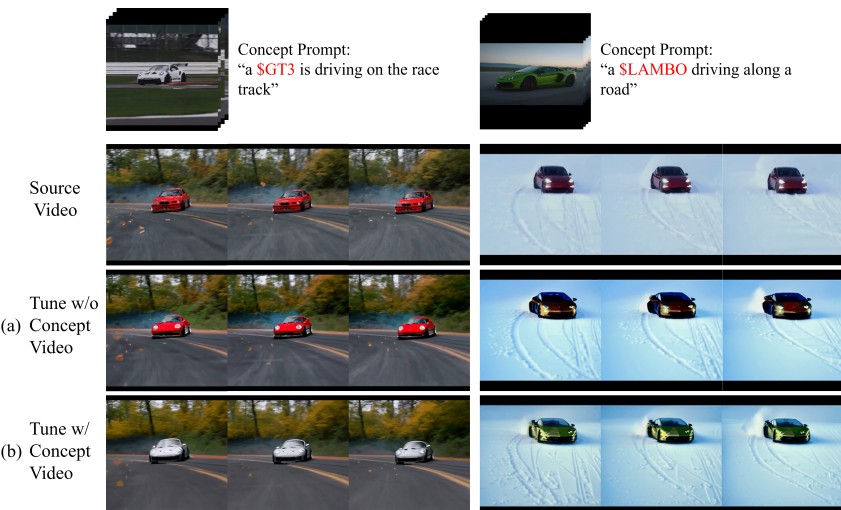

Figure 6: **Comparison of whether to tune with the concept video.** Compared the video editing results without and with tuning concept video for the left part: 'car' → '$GT3'; and the right part: 'car' → '$LAMBO', from the source prompt "a car is drifting around a curve road with the background of a forest" and "a car is drifting in the snow", respectively.

## 5 LIMITATIONS AND FUTURE WORK

**Mismatch when Significant Deformation.** Although our proposed method effectively mitigates the inconsistency in non-target areas caused by attention dispersion in video editing methods using attention replacement mechanisms, it may struggle when a single concept video guides target replacement in cases of significant deformation in the source video, such as running people. For instance, there may be insufficient detailed correspondences between the internal parts of the replacing and replaced targets during deformation, such as moving arms and legs. Potential solutions include ControlNet (Zhang et al., 2023) and OpenPose (Cao et al., 2019), which utilize motion conditions, like human pose, to guide the video editing process.

## 6 CONCLUSION

In this paper, we propose an improved concept-augmented video editing approach that generates diverse and stable target videos flexibly by devising abstract conceptual pairs. Specifically, the framework introduces a Concept-Augmented Textual Inversion (CATI), which extracts the target concept from user-customized videos. In practice, CATI enables plug-and-play guidance of stable diffusion for video shaping, effectively capturing target attributes for more stylized editing results. In addition, a Dual Prior Supervision (DPS) mechanism is designed to prevent unintended changes in non-target visual areas by crossing the attention between the sources and targets. Experimental results demonstrate that our method significantly improves the flexibility, consistency, and stability of text-guided video editing.

# 7 ETHICS STATEMENT

This research adheres to ethical guidelines and standards. No human subjects were involved, and the dataset used was publicly available and anonymized to ensure privacy. The methodologies employed were carefully chosen to avoid introducing bias or unfair outcomes. There are no conflicts of interest or sponsorships influencing the findings. All legal and institutional regulations, including IRB approval where necessary, were strictly followed.

# 8 REPRODUCIBILITY STATEMENT

This work provides a clear link to the anonymized code: https://anonymous.4open.science/r/STIVE-79D5/README.md. The details of the data used in the experiments have been clearly outlined in the main text, and additional results in the Appendix demonstrate convincing superiority.

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

# A  APPENDIX

## A.1  TRAINING AND INFERENCE DETAILS

**Details of Concept Augmented Textual Inversion.** During the stage of concept augmented textual inversion, we use AdamW optimizer (Loshchilov & Hutter, 2017), with the default betas set to 0.9 and 0.999, the weight decay set to $1 \times 10^{-2}$ and the epsilon set to $1 \times 10^{-8}$. Besides, LoRA modules inserted to UNet are without bias trainable parameters, in which the LoRA rank is set to 16, the weight coefficient to scale LoRA output is set to 1.0, and the dropout parameter is set to 0.1. Additionally, we add a prefix to the head of concept prompts and use one embedding to represent a concept word as the same as in Textual Inversion (Gal et al., 2022).

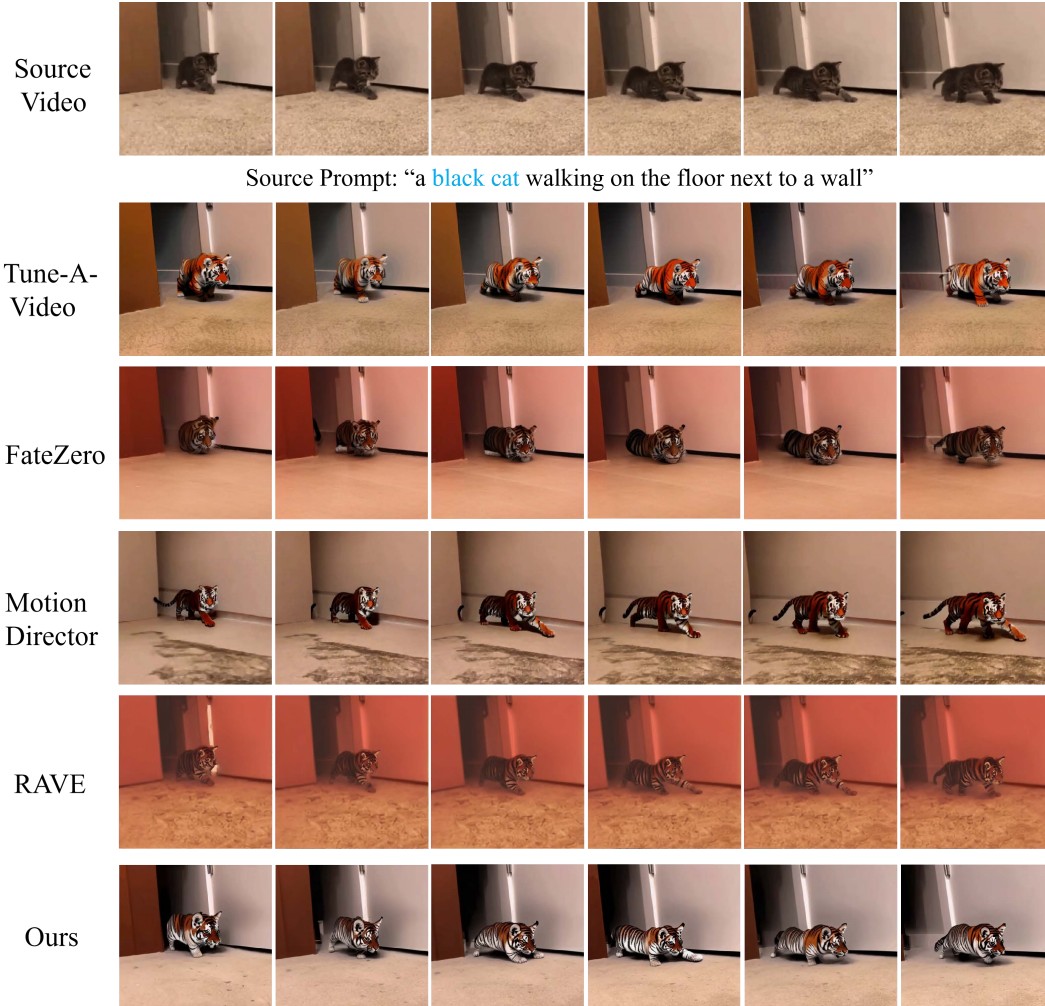

Figure 7: **Additional qualitative comparison of text-driven video editing without concept video.** At the top of the figure are the source video frames to be edited and its corresponding descriptive prompt. The rows below show the editing results of the source video using the editing prompt, for (Wu et al., 2023), (Qi et al., 2023), (Zhao et al., 2023), (Kara et al., 2024), and our method, respectively.

**Details of Dual Prior Supervision.** During the stage of tuning the temporally extended diffusion model with dual prior supervision, we use the same optimizer settings in concept augmented textual inversion stage. For each training epoch, we sequentially use the pairs of source video with source prompt and concept video with concept prompt as input for one training iteration, predicting noise

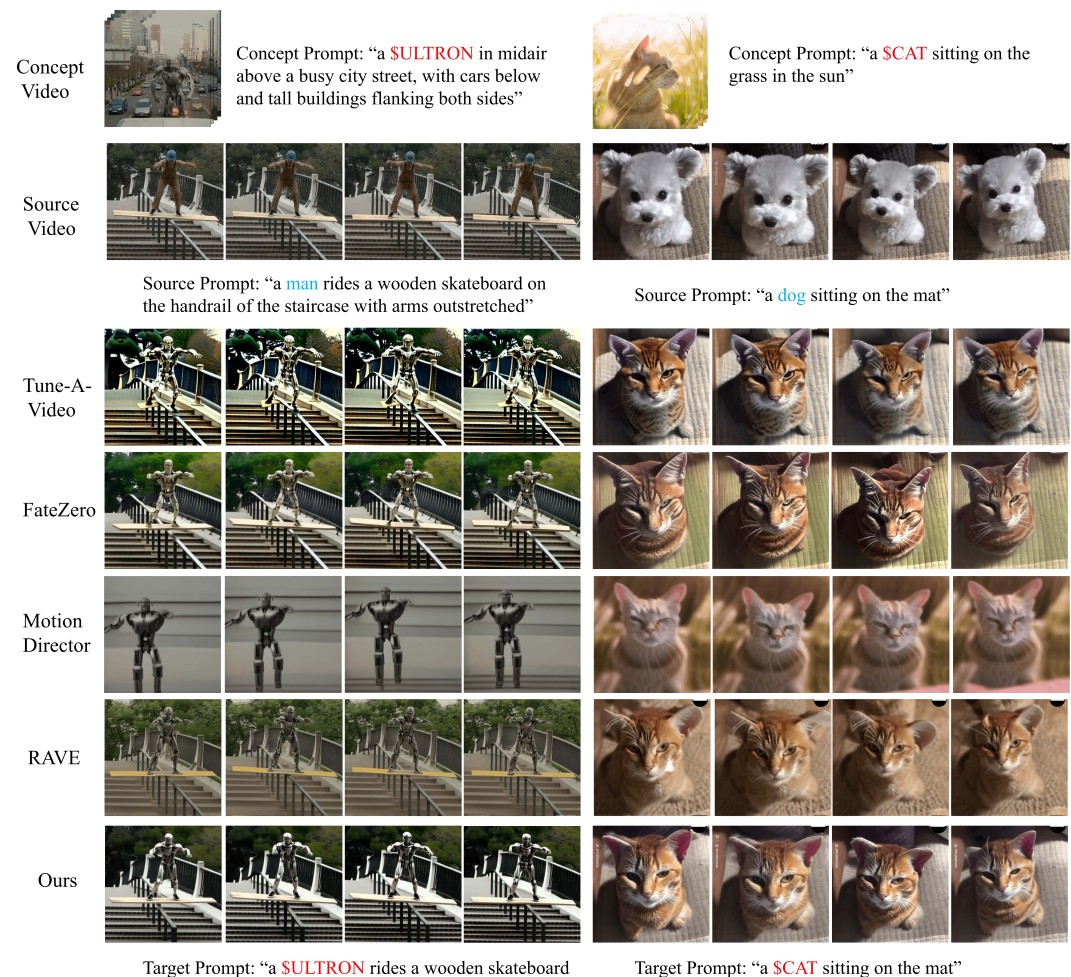

Figure 8: **Additional qualitative comparison of text-driven video editing with concept video.** At the top of the figure are the concept videos and the prompt containing the concept word. Below are the source video frames to be edited along with its descriptive prompt. The rows below show the editing results of the source video using the prompt containing the concept word, for (Wu et al., 2023), (Qi et al., 2023), (Zhao et al., 2023), (Kara et al., 2024), and our method, respectively.

and computing the noise prediction loss. During one training iteration, we retain all the cross-attention probability matrices from the cross attention layers in the UNet and compute *scam* based on the pseudo ground-truth mask corresponding to each video. Before the end of each iteration, we then input the pair of source video and target prompt, also retaining the cross-attention probability matrices, and compute *tcam* based on the pseudo ground-truth mask. The number of training epochs are less than 250. The input data frames have a resolution of $512 \times 512$ pixels and a length of 6 frames. The training time does not exceed 30 minutes on an NVIDIA GeForce RTX 4090 device.

**Details of editing w/ concept video in comparison methods.** The methods we compared (like Tune-A-Video (Wu et al., 2023), FateZero (Qi et al., 2023) and RAVE (Kara et al., 2024)) are all based on the same CLIP (Radford et al., 2021) text encoder that enables us to integrate the same input concept pairs into the models. For MotionDirector (Zhao et al., 2023), it supports a spatial path to bring the object of concept video into latent space originally. That is to say, we can make fair comparisons for existing approaches. Therefore, we are able to uniformly integrate the concept video into these existing methods, which ensures the fairness of our comparison experiments.

**Details of Inference.** During the stage of inference, we use the device equipped with an NVIDIA GeForce RTX 4090 and store the model parameters and data inputs in fp16 format. During the

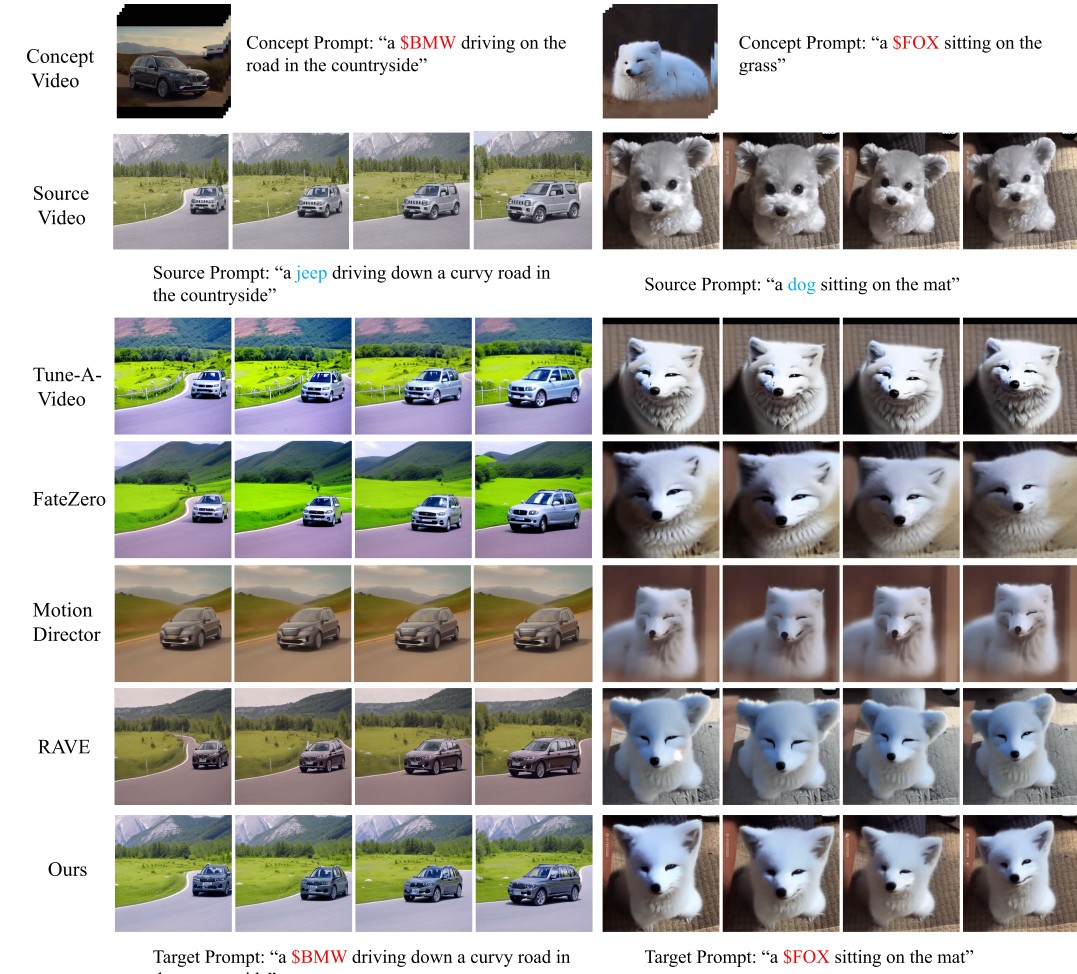

Figure 9: **Additional qualitative comparison of text-driven video editing with concept video.** The form is the same as Fig. 8.

denoising of the source video, we store intermediate variables including the self-attention and cross-attention probability matrices and source latents at each step in RAM, occupying approximately 100 GB space. The attention control processes of self-attention blending and cross-attention swap consume most of the inference time. Depending on different attention control configurations, the inference time generally ranges between 1 to 3 minutes.

Additionally, to reduce the host memory overhead of intermediate variables during inference, we have designed and implemented a memory-saving inference scheme. This scheme requires about twice the inference time compared to the original but reduces the overhead of RAM to 5 GB. The key difference between this scheme and the original is that it does not store the self-attention and cross-attention probability matrices; instead, it only stores the source latents at each denoising step and recalculates the self-attention and cross-attention probability matrices during attention control.

## A.2 ADDITIONAL COMPARISONS WITH EXISTING METHODS.

**Text-driven Video Editing without Concept Video.** We provide an additional set of comparison examples without concept video guidance for video editing, as shown in Fig. 7.

**Text-driven Video Editing with Concept Video.** We provide six additional comparison examples with concept video guidance for video editing, as shown in Fig. 8, Fig. 9, and Fig. 10.

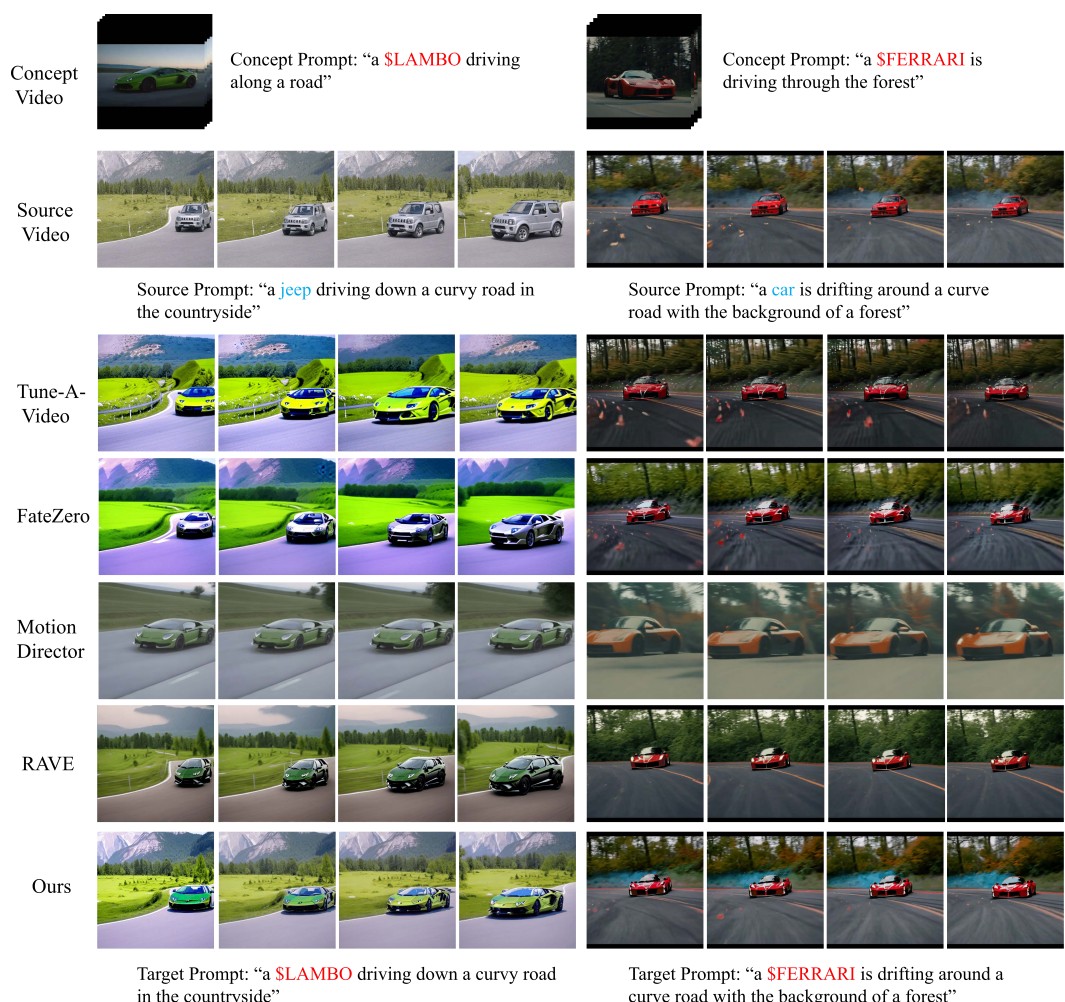

Figure 10: **Additional qualitative comparison of text-driven video editing with concept video.** The form is the same as Fig. 8.

## A.3 ADDITIONAL ABLATION STUDY RESULTS

**Additional Results for Dual Prior Supervision.** We provide the full comparison of the editing results for Fig. 5, as shown in Figure 11, and the comparison of the full average cross attention map in Fig. 12.

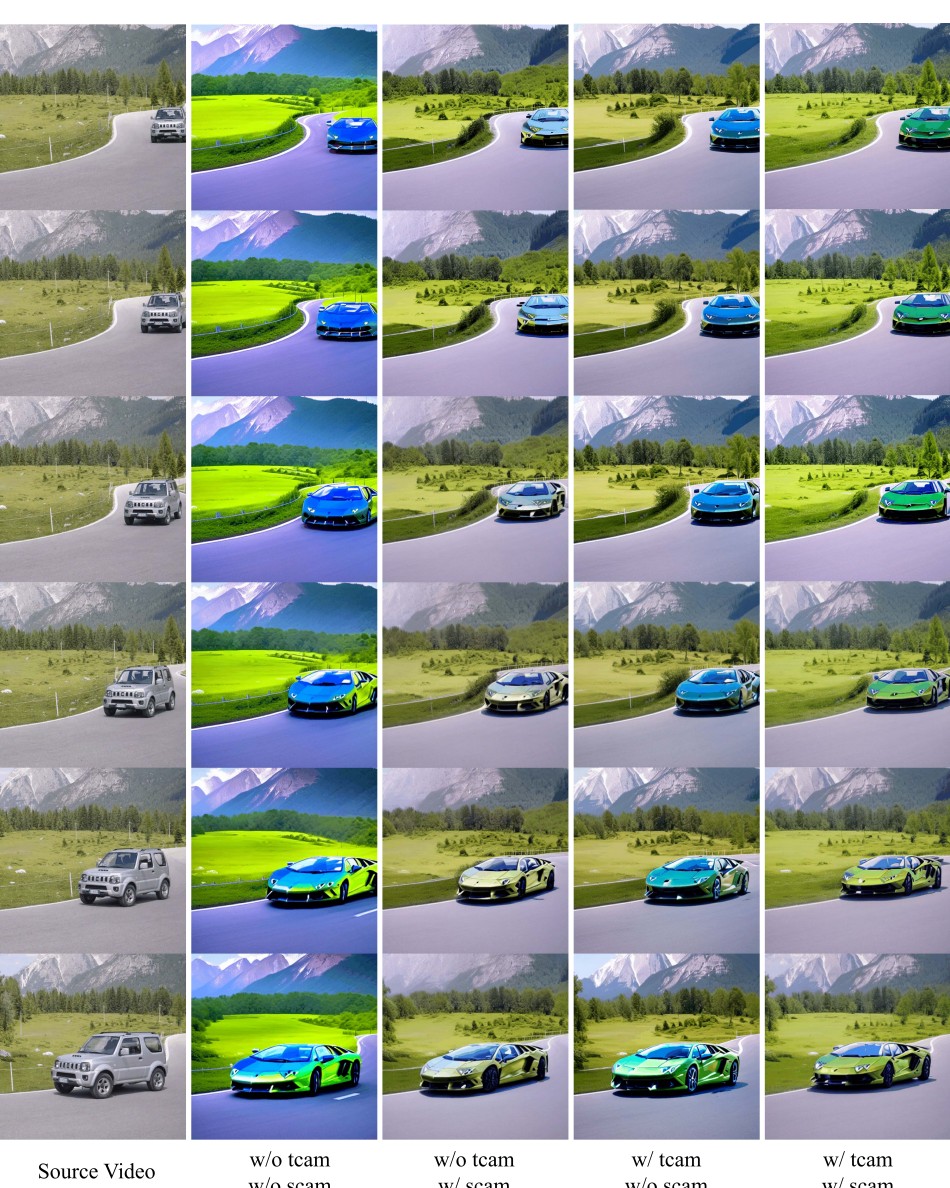

| Source Video | w/o tcam w/o scam | w/o tcam w/ scam | w/ tcam w/o scam | w/ tcam w/ scam |

Figure 11: **Full comparison of whether to tune with attention supervision.** The first column shows frames from the source video to be edited. From the second column to the last column, using the editing pair in Fig. 1 as the example, we compare the editing results with and without the attention supervision mechanism of scam and tcam.

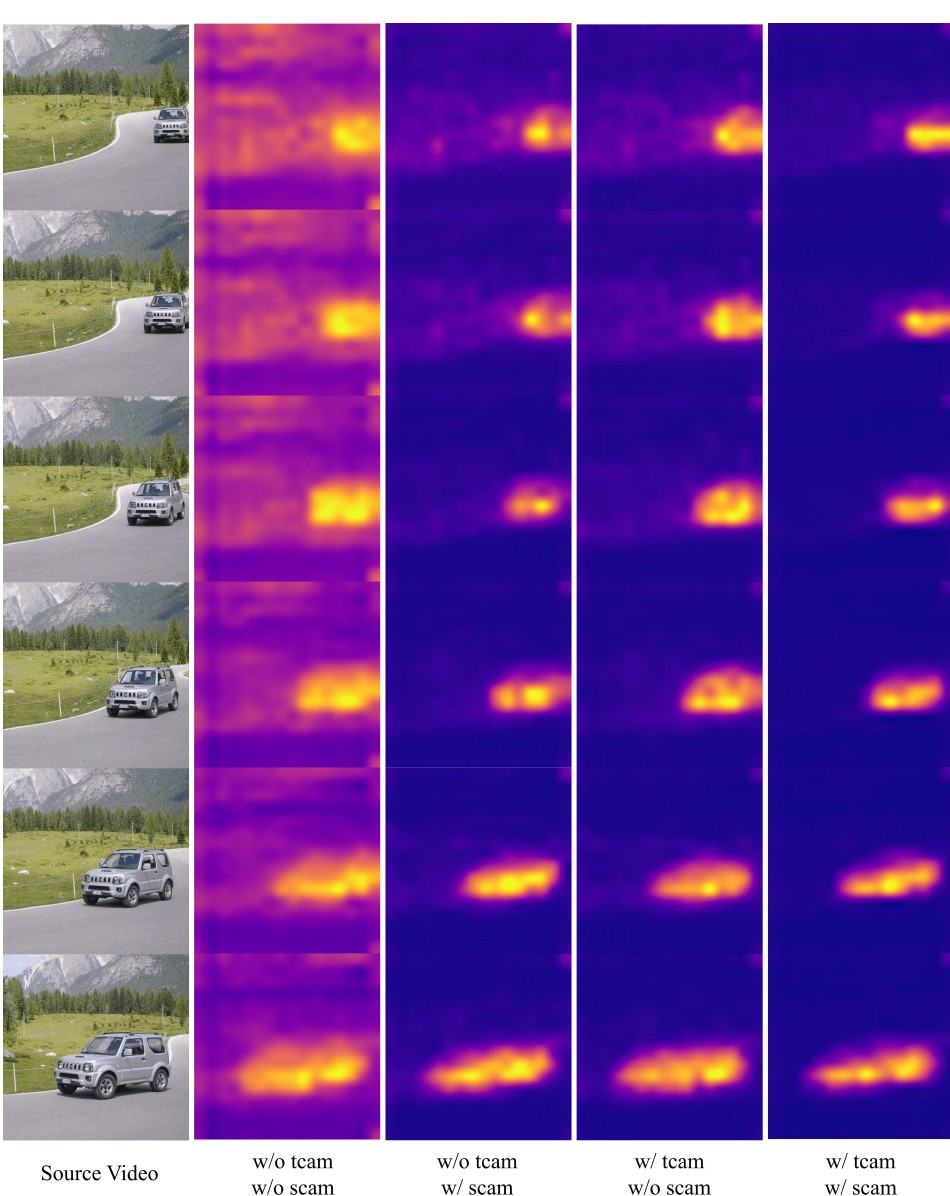

|  | w/o tcam | w/o tcam | w/ tcam | w/ tcam |
|--|----------|----------|---------|---------|
| Source Video | w/o scam | w/ scam | w/o scam | w/ scam |

Figure 12: **Average cross attention probability map explorations.** The first column shows frames from the source video to be edited. From the second column to the last column, using the editing pair in Fig. 1 as the example, we compare the average cross attention maps with and without the attention supervision mechanism of scam and tcam, visualization of plasma colormap in matplotlib (Hunter, 2007).

**Additional Results for Tuning w/ Concept Video Produces Stylized Results.** We provide additional comparisons of whether to tune with the concept video, as shown in Fig. 13.

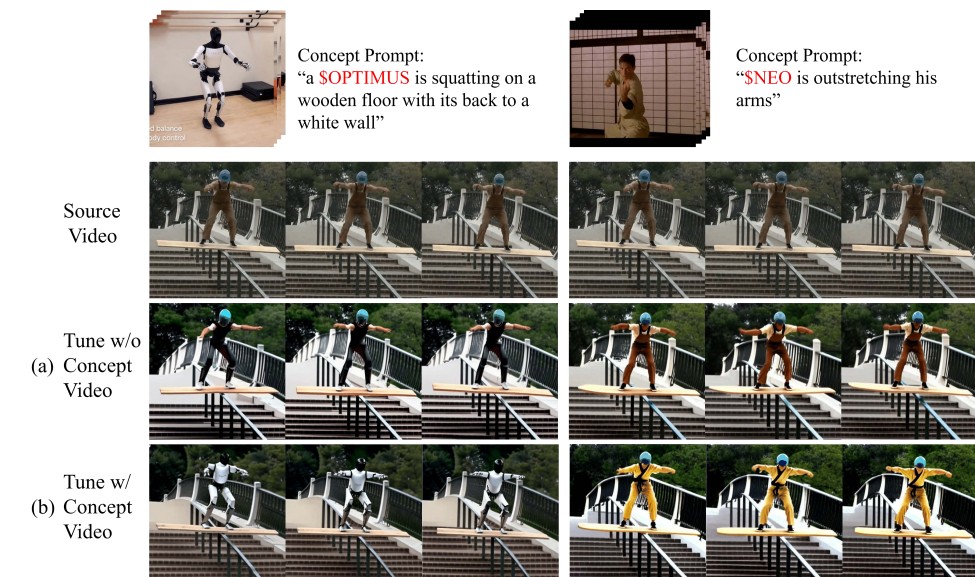

Figure 13: **Additional comparison of whether to tune with the concept video.** Compared the video editing results without and with tuning concept video for the left part: 'man' → '$OPTI-MUS'; and the right part: 'man' → '$NEO', from the same source prompt "a man rides a wooden skateboard on the handrail of the staircase with arms outstretched".

**Impact of Attention Supervision Weights.** Considering the impact of different attention supervision weights, Fig. 14 shows that as the weight increases, there is a higher degree of overlap between the area of the replacement target and the edited target. Simultaneously, the consistency in non-target areas improves to a certain extent.

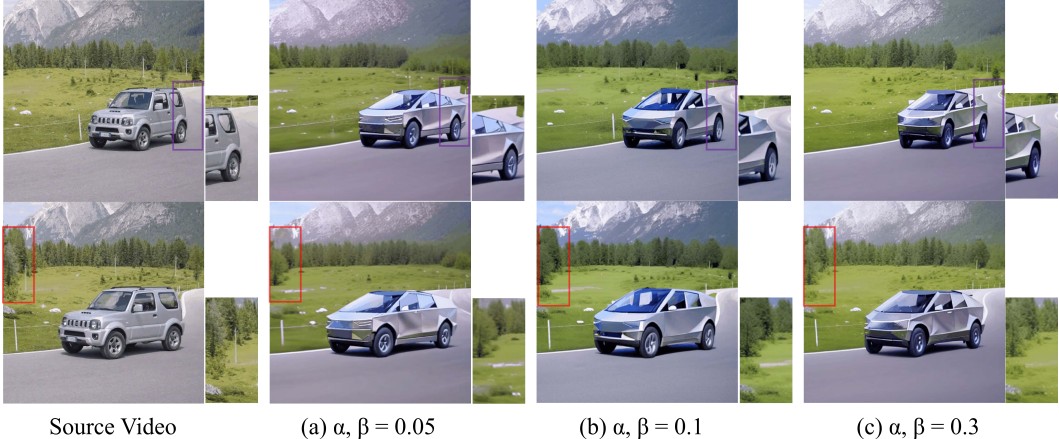

Source Video          (a) α, β = 0.05          (b) α, β = 0.1          (c) α, β = 0.3

Figure 14: **Hyperparameters of different attention supervision weights.** The first column shows frames from the source video to be edited. From the second column to the last column, using the editing pair in Fig. 4(c) as the example, we compare the editing results with attention supervision weights from 0.05, 0.1 and 0.3.

