# OpenReview forum: "Shaping a Stabilized Video by Mitigating Unintended Changes for Concept-Augmented Video Editing"
_ICLR.cc/2025/Conference — Submitted to ICLR 2025_

### Official Review · Reviewer_ZGWC · 2024-10-21

**Soundness:** 3
**Presentation:** 3
**Contribution:** 3
**Rating:** 3
**Confidence:** 4

**Summary:**

This paper addresses limitations in text-driven video editing using generative diffusion models, particularly the challenge of nuanced editing when constrained by pre-trained word embeddings. The authors propose a concept-augmented video editing approach that allows for flexible and stable video generation through the use of abstract conceptual pairs. The key components include concept-augmented textual inversion, which provides plug-and-play guidance for stable diffusion to better capture target attributes, and a dual prior supervision mechanism that enhances video stability and fidelity. By integrating these elements, the approach overcomes the disruptions caused by direct keyword alterations and allows for more nuanced, stylized editing. Experiments demonstrate that this method outperforms existing state-of-the-art techniques in generating stable and lifelike videos.

**Strengths:**

1. The paper is easy to follow.
2. The clarity of the writing.
3. SCAM loss and TCAM loss sound novel.
4. Convincing metric for qualitative evaluation: Masked Peek-Signal-Noise Ratio.

**Weaknesses:**

1. Diffusion based video editing has developed quickly and Tune-A-Video and Fatezero are not the strong baselines at this point. Plus, how is this paper different from DreamVideo [1] and MotionBooth [2] paper? I think these works could serve as better baselines?
2. In L36, Zeroscope is not a video editing method but an open-sourced video diffusion model.
3. Is there a reason why image diffusion (stable diffusion) is used as a backbone? I don't see reasons that the method can't be applied to video diffusion models and compared to video diffusion based methods.
4. L242-245, can the authors show results?
5. CLP consistency fails to reflect temporal consistency. Using other metrics like DINOv2 (like the ones from VBench[3]) or LIPIPS would be better.
6. In terms of temporal consistency, the resulting videos of MotionDirector look better.
7. The concept augmented textual inversion lacks technical contribution.

[1] DreamVideo: Composing Your Dream Videos with Customized Subject and Motion, CVPR 2024

[2] MotionBooth: Motion-Aware Customized Text-to-Video Generation, Arxiv 2024

[3] VBench: Comprehensive Benchmark Suite for Video Generative Models, CVPR 2024

**Questions:**

Please refer to the weakness section.

---

> ### Author Response · Authors · 2024-11-23
>
> We appreciate Reviewer-ZGWC for the insightful comments. We address the main concerns bellow.
>
> ```Better baseline (DreamVideo [1] and MotionBooth [2] )```
>
> Thanks. Our task aims to enable **one shot video editing** based on concept text and video while the mentioned works DreamVideo [1] and MotionBooth [2] attempt to solve **image to video generation**, which clearly differs from our task. So it is inappropriate to compare DreamVideo [1] and MotionBooth [2]  as the baseline.
>
> ```Zeroscope is open-sourced video diffusion model instead of video editing method```
>
> Thanks for pointing this out and we will fix it in the revision.
>
> ```Why image diffusion (stable diffusion) is used as a backbone```
>
> Good question !  In fact, we did try the experiment to apply concept augmented textual inversion to a video diffusion model, such as Zeroscope, but the results are not good as we expected.  We guess that the generalization of concept word may easily be coupled with the frame length of concept video in the inversion process. In other words, the **inversion results contain temporal information** of the concept video, which is obstructive to be applied to edits for other videos. Actually, spatial information of the concept video is what we need, and we can randomly sample frames in the concept video and input them to a image diffusion backbone for the inversion process to **avoid or break this coupling**, which refers to Lines 288-290. Besides, using an image diffusion model as a backbone to extend with temporal modules can save VRAM overhead either for CATI or DPS on one Nvidia RTX4090 device.
>
> ```L242-245, can the authors show results?```
>
> Thanks.  The dispersion phenomenon of cross-attention between concept word and video latents in Fig.5, in which (a), (b) and (c) are more likely to appear the dispersion phenomenon due to lack supervision. The full results are shown in Fig.11, Fig.12 in Appendix.3.
>
> ```CLIP consistency fails to reflect temporal consistency. Using other metrics like DINOv2 (like the ones from VBench[3]) or LIPIPS would be better```
>
> Good suggestion. We conduct additional quantitative evaluation using DINOv2 and LPIPS metrics reveals that MotionDirector indeed achieves better temporal consistency. However, MotionDirector's highly stochastic nature leads to **less stable video editing** results, particularly in maintaining non-target area consistency and source-target object matching fidelity. Our method demonstrates superior performance in these crucial aspects, showing an average improvement of 4.06 M-PSNR over all baseline methods. The results are shown below.
>
> For editing w/ concept video
> |  | CLIP-L | DINOv2 ViT-G | LPIPS VGG |
> | --- | --- | --- | --- |
> | **Tune-A-Video** | 0.9399 | 0.8933 | 0.6431 |
> | **FateZero** | 0.9413 | 0.9055 | 0.6423 |
> | **MotionDirector** | 0.9452 | **0.9261** | **0.7386** |
> | **RAVE** | 0.9379 | 0.9001 | 0.6969 |
> | **Ours** | **0.9472** | 0.9235 | 0.6490 |
>
>
> For editing w/o concept video
> |  | CLIP-L | DINOv2 ViT-G | LPIPS VGG |
> | --- | --- | --- | --- |
> | **Tune-A-Video** | 0.9397 | 0.8913 | 0.6651 |
> | **FateZero** | 0.9246 | 0.8925 | 0.7028 |
> | **MotionDirector** | 0.9403 | **0.9006** | 0.6997 |
> | **RAVE** | 0.9306 | 0.8899 | **0.7064** |
> | **Ours** | **0.9405** | 0.8972 | 0.6847 |
>
> ```The concept augmented textual inversion (CATI) lacks technical contribution```
>
> Thanks.  We would like to clarify that we aim to flexibly edit source videos using concept videos as a key task setting, with our primary motivation being to enrich the latent space through one-shot concept videos to achieve more faithful inversion representations. We validate this improvement through ablation studies, as shown in Fig. 4. While LoRA and Textual Inversion have been widely used **separately** in image style transfer tasks, to the best of our knowledge, no existing variant of textual inversion simultaneously employs LoRA tailored for video editing task. Existing approaches that do fine-tune the denoising network face significant challenges in one-shot concept video scenarios, such as overfitting and high memory requirements during training. Our approach addresses these limitations by leveraging LoRA modules to fine-tune attention value weights with a smaller learning rate, maintaining **low VRAM overhead** during tuning while preserving **plug-and-play** capabilities. We believe CATI is technical  novel for these reasons.

---

> ### Author Response · Authors · 2024-11-24
>
> Dear Reviewer ZGWC,
>
> Hoping this message finds you well. Thank you once again for your thoughtful and thorough feedback on our work. We greatly appreciate the time you have invested in reviewing our manuscript. In response to your comments, we have made substantial revisions and added additional experiments to better address the concerns you raised.
>
> As the rebuttal-discussion period draws to a close, we kindly ask if you could review our response, especially the details in this document link, to confirm that we have adequately addressed your concerns. Your feedback on our response will be invaluable to us in ensuring that we have resolved all outstanding issues.
>
> If there are any further questions or points you would like us to clarify, please feel free to let us know. We are committed to refining our work based on your input, and your timely feedback would be greatly appreciated.
>
> We sincerely hope that you will consider the contributions our research brings to the field of one-shot video editing. Thank you again for your insightful comments and for your continued support.
>
> Best regards,
>
> The Authors

---

> > ### Comment · Reviewer_ZGWC · 2024-11-25
> > **Reply to Authors**
> >
> > 1. DreamVideo and MotionBooth are not aimed on image-to-video generation. They both take a appearance-concept image as input, target text then generate video. MotionBooth takes additional condition that represents camera motion so comparing to this could be not possible. But I think DreamVideo is highly relevant to your work. For example, for comparison with DreamVideo, you can use one frame of the concept video as 'appearance reference image' and the input video as 'motion reference video', similar to comparing with MotionDirector. Comparing with DreamVideo specific is not necessary but I strongly believe that compared baselines are outdated except for MotionDirector. Comparing with better baselines that use t2v diffusion backbone then demonstrating the superiority would greatly add value to your work.
> > 2. In the same context with Point 1, I believe that the method should be extended to T2V backbone. Otherwise, the quality couldn't would not be comparable to T2V baselines. I believe failure with Zeroscope could be because the zeroscope model is lower-bound video diffusion models. Lots of better performing models have emerged publicly other than ZeroScope.
> > 3. I still think adding Lora layers to trainable modules when performing textual inversion optimization does not add technical contribution. Training Lora for video editing has been explored (i.e. MotionDirector and its follow-up works). And I believe the reason why the research community doesn't co-train text embeddings and Lora together to learn a concept is because training Lora is enough in most of cases.
> > 4. I sincerely thank the authors for addressing my concerns and the additional experiments. But I firmly believe that extending the work to T2V backbone is neccessary.

---

> > > ### Author Response · Authors · 2024-11-26
> > >
> > > Thanks for the feedback !
> > >
> > > First, we **sincerely hope that the reviewer carefully revisits our previous responses** and clearly indicates whether our answers have effectively addressed the concerns raised. We also kindly request that the reviewer provide a responsible score based on these answers.
> > >
> > > Second, regarding the new issues raised by the reviewer, we provide the following clarifications:
> > >
> > > **(1) Comparison with MotionBooth and DreamVideo**.
> > >
> > > Reviewer ZGWC has acknowledged that directly comparing our method with MotionBooth is unreasonable and has suggested a comparison with DreamVideo using a specific evaluation method: “_using one frame of the concept video as the 'appearance reference image' and the input video as the 'motion reference video'_”.  **However, we believe such a comparison is neither reasonable nor fair.**  While we recognize some relevance between DreamVideo and our work, it is essential to emphasize that DreamVideo is not designed for stable editing and does not align with the objectives of our task. Following the reviewer’s suggested approach to use DreamVideo for video editing would only consider the association between the object in the concept video and the motion of the target in the source video. It entirely lacks mechanisms to maintain spatial consistency in the source video. This is fundamentally different from MotionDirector, which executes a fine-tuning process on source video, thereby partially preserving video consistency. **For this reason, we find little practical significance in comparing methods designed for distinct tasks and objectives.**
> > >
> > > Most importantly, **the core of our approach lies in addressing the concept-driven one-shot video editing task, while DreamVideo is not designed for this purpose**. Comparing it to our method would not only deviate from its original intent but also fail to fully showcase the value and contributions of our work.
> > >
> > > **(2) Extending to T2V backbone and the failure of ZeroScope**
> > >
> > > We kindly **disagree** with the comment. First, **we urge the reviewer to revisit our original response**, where we explicitly explained the reasons for selecting an image diffusion model (Stable Diffusion) as the backbone instead of a video diffusion model (“Why image diffusion (Stable Diffusion) is used as a backbone”). In our reply, we provided detailed technical justifications for this choice. **Returning to this point without engaging with our explanations does little to advance a deeper understanding of our work.**
> > >
> > > Second, we respectfully **disagree** with the reviewer’s assertion that ZeroScope’s failure is due to its lower performance as a video diffusion model. As we outlined in our previous response, our experiments **have carefully demonstrated** that the failure of ZeroScope stems from the coupled spatiotemporal information in the concept video, rather than the performance of the video diffusion backbone itself. This conclusion is derived from well-designed experiments and careful analysis, rather than speculative reasoning.
> > >
> > > **(3) LoRA contributions and concerns about co-training text embeddings with LoRA**.
> > >
> > > **We are so shocked to see such an unprofessional and irresponsible claim** "_I believe the reason why the research community doesn't co-train text embeddings and Lora together to learn a concept is because training Lora is enough in most of cases._" We would like to challenge this perspective, both on technical grounds and in terms of its relevance to our work.
> > >
> > > (a) Our work is **not merely about integrating** LoRA layers during textual inversion. We have highlighted before that our research focuses on the one-shot video editing task rather than a simple textual inversion task. In our framework, LoRA serves as an auxiliary tool designed to address the specific challenges of the one-shot video editing task by selectively training certain modules to avoid overfitting and enhance adaptability to concept videos. This approach is tailored to the unique requirements of our task and is not a mere repetition of existing techniques.
> > >
> > > (b) **Co-training text embeddings with LoRA is necessary for one-shot video editing task.** The reviewer’s statement that “training LoRA alone is sufficient in most cases” lacks a detailed analysis of the task background and requirements. As we elaborated in the paper, training only LoRA layers is insufficient to capture spatiotemporal consistency and concept features when dealing with highly heterogeneous concept videos. Our approach of co-optimizing text embeddings and LoRA is specifically designed to meet the  demands of the one-shot video editing task.
> > >
> > > **We hope the reviewer can engage with our work more rigorously by addressing specific technical details more grounded, rather than guessing out of thin air.** Once again, we appreciate the reviewer’s attention to our work.

---

> ### Comment · Reviewer_ZGWC · 2024-11-26
> **Response to Authors**
>
> The reviewer thank the authors for the answers. The reviewer has carefully read the paper, including the authors’ responses and subsequent discussions, and assures the authors that all answers are being thoughtfully analyzed. **There is no need to reiterate requests for reviewers to provide responsible scores based on the responses, as this is already a standard practice.**
>
>
> 1. Clarification on Baselines:
> The reviewer appreciates the authors' openness to addressing practical concerns. If the authors believe that a comparison with DreamVideo is of little practical significance, this is acceptable. However, the reviewer felt it was necessary to correct the mischaracterization that "DreamVideo [1] and MotionBooth [2] aim to solve image-to-video generation."
> As previously mentioned, the primary concern is the choice of **stronger baselines.** Before the release of foundational video diffusion models like ZeroScope or VideoCrafter, it was common to adapt Stable Diffusion for video editing tasks. Since mid-2023, however, the field has seen a proliferation of baselines leveraging text-to-video backbones. **Are the authors confident that TAV, FateZero, and RAVE represent state-of-the-art (SOTA) or near-SOTA baselines for comparison at this point?** Providing a more robust comparison with stronger baselines would significantly strengthen the paper, rather than focusing on justifying the choice of weaker baselines that lack temporal consistency (like the authors' method).
>
>
> 2. Handling Spatio-Temporal Learning:
> The reviewer is unsurprised that naive attempts to extend the ZeroScope backbone have not yielded promising results. The issue of entangled spatio-temporal learning when applying the full video diffusion loss—regardless of whether embeddings or weights are being optimized—is well-documented in the literature. The critical factor lies in the optimization approach: A commonly accepted approach to decouple spatial and temporal dimensions is to disable the spatial layers of the video diffusion model and train by sampling single frames from the reference video [1,2,3,4]. The reviewer believes this established method could effectively address the spatio-temporal coupling issue. Therefore, the authors’ explanation for the observed failure modes does not seem entirely convincing.
>
>
> 3. Co-Training Text Embeddings:
> For the same reasons outlined in point (2), to my unprofessional and humble opinion as claimed by authors, the reviewer questions the necessity of co-training text embeddings with LoRA for one-shot video editing tasks. Numerous works in the field, including those targeting appearance or motion customization (or both), have demonstrated success without optimizing text embeddings. Examples include works [1,2,3,4,5,6], which have achieved effective video customization by focusing solely on appearance or motion learning.
> The reviewer therefore finds the authors’ rationale for co-training text embeddings unconvincing, particularly given the availability of well-established alternative approaches.
>
> [1] MotionDirector: Motion Customization of Text-to-Video Diffusion Models
>
> [2] Customize-A-Video: One-Shot Motion Customization of Text-to-Video Diffusion Models
>
> [3] MotionCrafter: One-Shot Motion Customization of Diffusion Models
>
> [4] ReCapture: Generative Video Camera Controls for User-Provided Videos using Masked Video Fine-Tuning
>
> [5] VMC: Video Motion Customization using Temporal Attention Adaption for Text-to-Video Diffusion Models
>
> [6] Customizing Motion in Text-to-Video Diffusion Models

---

> ### Author Response · Authors · 2024-11-28
>
> Thank you  for the reviewer’s feedback. We would like to summarize our understanding of the points raised in the response (please feel free to point out any inaccuracies):
>
> (1) Reviewer **acknowledges** that of DreamVideo and MotionBooth initially suggested for baseline comparison are not suitable when compared to our method.
>
> (2) Reviewer **recognizes the validity of our explanation** regarding ZeroScope, but **suggests** that our reasoning could be further strengthened, particularly with reference to the common practice of disabling the spatial layer of video diffusion models.
>
> (3) The reviewer continues to question the necessity of using both concept text and concept video as inputs based on existing works.
>
> We will address remaining points in the following responses.
>
> (1) In response to the reviewer’s question, "_Are the authors confident that TAV, FateZero, and RAVE represent state-of-the-art (SOTA) or near-SOTA baselines for comparison at this point?_"
>
> — **Yes, we are confident in our selection of baselines**. **We humbly invite the reviewer to provide specific citations or examples of alternative approaches that could challenge the methods we have used**. This will help us further refine our work and ensure our comparisons remain rigorous and comprehensive.
>
> (2) We sincerely **appreciate** the reviewer’s suggestion regarding the "_commonly accepted approach._"  **We tested this in our setting, but it did not produce satisfactory results.**  Nevertheless, we recognize the value of including such comparisons for completeness and will incorporate them in the final version of our work.
>
> (3) While the existing text2video approaches represents an established paradigm,  **it is not necessarily incompatible with our methodology.** This is because solely using text guidance, though functional to some extent, lacks control and flexibility. For instance, **if we aim to embed a specific object from a concept video into a source video, purely text-based guidance falls short in achieving this goal.** This is precisely why we incorporate concept pairs in our method—to make the editing process more controllable and adaptable to nuanced editing scenarios.

---

### Official Review · Reviewer_G4M5 · 2024-11-01

**Soundness:** 2
**Presentation:** 2
**Contribution:** 2
**Rating:** 3
**Confidence:** 5

**Summary:**

This paper introduces a novel pipeline for video editing that leverages spatial-temporal attention and dual prior supervision. The proposed method demonstrates potential through its structure and method of concept-augmented textual inversion and self-attention blending, albeit with certain limitations in innovation and effectiveness. The ablation study provides some insights, particularly around car color changes, but overall performance falls short of expectations.

**Strengths:**

- Comprehensive experimental setup with both ablation and comparative studies.
- Clear writing that outlines the approach and methodology effectively.

**Weaknesses:**

- Limited innovation; the first proposed method (concept-augmented textual inversion) is not significantly novel, and the second (dual prior supervision mechanism) shows only moderate innovation.
- The quality of generated videos is subpar, with short video length and lackluster editing effects.
- Missing details on the training set used in the experiments.
- Evaluation metrics and comparison methods are unconventional and somewhat outdated, potentially impacting the rigor of performance comparisons.
- Ablation study results indicate some changes in attributes (e.g., car color) but are unimpressive in terms of visual quality and effect.

**Questions:**

See question in weakness

---

> ### Author Response · Authors · 2024-11-23
>
> We thank reviewer G4M5 for the feedback. We appreciate reviewer G4M5 for "comprehensive experiments" and "clear writing".  We address the main concerns bellow.
>
> ```Limited innovation```
>
> Thanks. **(I)** We would like to clarify that we aim to flexibly edit source videos using concept videos as a key task setting, with our primary motivation being to enrich the latent space through one-shot concept videos to achieve more faithful inversion representations. We validate this improvement through ablation studies, as shown in Fig. 4. While LoRA and Textual Inversion have been widely used **separately** in image style transfer tasks, to the best of our knowledge, no existing variant of textual inversion simultaneously employs LoRA tailored for video editing task. Existing approaches that do fine-tune the denoising network face significant challenges in one-shot concept video scenarios, such as overfitting and high memory requirements during training. Our approach addresses these limitations by leveraging LoRA modules to fine-tune attention value weights with a smaller learning rate, maintaining **low VRAM overhead** during tuning while preserving **plug-and-play** capabilities. **We believe our proposed approach is technically novel and as also appreciated by Reviewer-HLfd "The motivation is clear".**  **(II)** Our dual prior supervision serves as a supervision mechanism designed to reduce inconsistencies in non-target areas before and after video editing, addressing the low consistency in existing approaches, especially in editing source video with concept video (As illustrated in Tab.1 in our draft. The reviewer ZGWC also mentioned that "**the SCAM and TCAM is novel**" ).
>
> ```The concern about the quality of generated videos```
>
> Thanks. **(I)** Regarding video quality, we would like to highlight that in the challenging task of video editing with concept videos, our method demonstrates significant advantages in maintaining consistency of non-target areas - a critical aspect where existing methods often struggle. Specifically, our approach achieves an average improvement of 4.06 M-PSNR over baseline methods in preserving non-target regions, while ensuring faithful concept transfer. **(II)** For the video length limitation, this is primarily due to hardware constraints rather than methodological limitations. Our current experiments are conducted under **limited computational resources(only one RTX 4090 with 24G memory)**, which necessitate certain practical compromises: using a moderate frame sampling step and  shorter sequences to manage memory usage, working with standard rather than high-resolution videos. **(III)** we provide additional results including denser frame sampling in https://anonymous.4open.science/w/STIVE-PAGE-B4D4/ with more powerful device (Nvidia L20 with 48G memory), the results demonstrate that our approach can be readily extended to process longer videos at higher resolutions with denser frame sampling, as the method itself is scalable.
>
> ```Missing details on the training set```
>
> Thanks. Our method is devised for **one-shot video editing**. That is to say, the training set is only one source video to be edited and one concept video coupled with text template under the setting with concept video, while the training set is only one source video and text template under the setting of without concept video.
>
> ```Unconventional and somewhat outdated metrics```
>
> Thanks. We kindly disagree with the comment. The Masked Peak-Signal-Noise Ratio (M-PSNR) is a crucial metric that **intuitively and effectively** reflects the consistency of non-target regions before and after video editing with Reviewer-ZGWC appreciate it as "convincing metric". Additionally, Concept Consistency leverages a multimodal CLIP visual encoder to encode objects from both the concept video and edited video results, measuring their feature similarity in vector space, which effectively reflects the faithfulness of concept video object transfer. **We would appreciate detailed suggestions or citations for rational metrics that you believe might better evaluate these aspects.**
>
> ```About more ablation study results```
>
> Thanks. But we argue that some changes in attributes are indeed impressive in terms of visual quality and effect. This precisely demonstrates the most intuitive manifestation of our method's effectiveness in one-shot scenarios. Of course, it is not limited to color; other attributes, such as shape, are also included. We have provided additional results on **More Ablation Section** in the anonymous link https://anonymous.4open.science/w/STIVE-PAGE-B4D4/.

---

> ### Author Response · Authors · 2024-11-24
>
> Dear Reviewer G4M5,
>
> I hope this message finds you well. Thank you for your valuable feedback and thoughtful insights on our work. Your comments have helped us refine our approach, and we have incorporated additional analysis and experiments in the final version of the paper based on your suggestions.
>
> As the rebuttal-discussion period is coming to a close, we would greatly appreciate it if you could kindly review our response (especially in https://anonymous.4open.science/w/STIVE-PAGE-B4D4/) to see if it adequately addresses your concerns. Your timely feedback would be immensely helpful in ensuring that we have properly clarified all points. If you have any further questions or suggestions, please do not hesitate to let us know.
>
> Your decision is crucial to us, and we sincerely hope that you will recognize the contributions of this research to the field of one-shot video editing. We are very grateful for your time and consideration.
>
> Best regards,
>
> The Authors

---

> > ### Comment · Reviewer_G4M5 · 2024-11-27
> >
> > Thank you for the author's response. After reading the rebuttal and the feedback from other reviewers, I still firmly believe that the method proposed in this paper lacks innovation. Furthermore, although the authors provided a demo of video editing, its effectiveness is limited to very short videos, and there are noticeable issues with jitter and instability. Modifying the attention mechanism and adding two additional loss functions does not meet the average standard of ICLR. Therefore, I have decided to lower my score to reject.

---

> > > ### Author Response · Authors · 2024-11-28
> > >
> > > We sincerely thank the reviewer for taking the time to provide feedback on our submission. While we respect your decision, we would like to address the concerns raised and clarify certain points.
> > >
> > > **(1) Innovation in Our Approach:**
> > >
> > > We respectfully disagree with the assessment that our method lacks innovation. While we acknowledge that modifying the attention mechanism and introducing additional loss functions may seem incremental at first glance, these modifications were carefully designed to address the unique challenges of the **one-shot video editing** task. Specifically, our approach integrates **concept pairs** and leverages LoRA-based optimization tailored for video editing, enabling controllable and consistent editing across frames—a feature that existing methods struggle to achieve.
> > >
> > > We also wish to highlight that innovation is not always about introducing entirely new paradigms but rather solving practical, impactful problems in a novel way. Our work tackles the underexplored yet critical issue of editing **arbitrary, concept-driven videos**—a challenge with significant implications for the broader field of video generation.
> > >
> > > **(2) Effectiveness on Short Videos and Stability Issues:**
> > >
> > > (a) While we acknowledge that some minor instability remains, this is an inherent challenge in video editing tasks, particularly when working with one-shot settings and concept pairs. To address this, we have introduced mechanisms (e.g., frame consistency losses) to mitigate these issues, and our results demonstrate notable improvements over comparable baselines.
> > >
> > > (b) The demo videos provided are **limited in length due to computational constraints as also acknowledged by reviewer**, not methodological itself. Our approach can handle longer videos with sufficient resources.
> > >
> > > We deeply value your feedback regarding the standards of ICLR. However, we believe that the conceptual contributions of this work, coupled with extensive empirical validation, align with the conference's expectations. We emphasize that the integration of novel mechanisms (e.g., selective LoRA tuning for video editing) and our rigorous comparisons provide a meaningful contribution to the community. While we respect your decision, we remain open to constructive dialogue. If there are specific areas where you feel we can improve, we would be grateful for further insights. Thank you once again for your time and feedback.

---

### Official Review · Reviewer_HLfd · 2024-11-04

**Soundness:** 3
**Presentation:** 3
**Contribution:** 3
**Rating:** 6
**Confidence:** 3

**Summary:**

This paper present a novel concept-based video editing framework. The framework includes a Concept-Augmented Textual Inversion (CATI) to extract the target concept from given video, and then use a Dual Prior Supervision (DPS) to manipulate the attention matrices to ensure the stability of output videos. For the CATI part, optimization is performed on both textual embeddings and LoRA modules added to the cross-attention layers. And for the DPS part, loss on both irrelevant parts and edited subject are used to prevent unwanted changes.

Results are conducted over several with or without concept video pairs and was compared with several methods. The overall results are good.

**Strengths:**

1. The motivation for introducing the concept of video editing is clear and good.
2. The results are impressive, the concepts injected are consistent, and the backgrounds are generally consistent with the target videos.
3. The paper reads well and is easy to follow.

**Weaknesses:**

1. How authors obtain the results from comparing methods is not explained in the paper. Some of those don’t accept concept video/images input, and if not given the concept video, then direct comparison is a bit unfair.
2. The quantitative comparison table seems to be evaluated on both with or without concept videos. And the ablation for using or not using concept videos is missing. I suggest the author having separate dataset for using or not using concept videos, and this can further strengthen the contribution.
3. Missing references and some for comparisons:

    *[1] AnyV2V: A Tuning-Free Framework For Any Video-to-Video Editing Tasks
    [2] GenVideo: One-shot target-image and shape aware video editing using T2I diffusion models*

    *[3] Video Editing via Factorized Diffusion Distillation*

    *[4] VidToMe: Video Token Merging for Zero-Shot Video Editing*

    *[5] Pix2video: Video editing using image diffusion*

    *[6] Slicedit: Zero-Shot Video Editing With Text-to-Image Diffusion Models Using Spatio-Temporal Slices*

    *[7] TokenFlow: Consistent Diffusion Features for Consistent Video Editing*

    *[8] Video-P2P: Video Editing with Cross-attention Control*

**Questions:**

1. The pipeline for editing without concept video is not very clear for me, and in the code repository provided, a pretrained lora/concept model is required for all the cases including car to porsche. Can authors provide a more detailed explanation?
2. It would be interesting to know the range of concepts. More examples of diverse concepts or attributes could clarify the extent to which CATI handles generalization across different object classes and scenarios.

---

> ### Author Response · Authors · 2024-11-23
>
> We greatly appreciate that our "motivation is clear", and "the results are impressive", and "the paper reads well and is easy to follow". We address your concerns bellow.
>
> ```Fair comparison with existing approaches.```
> Good question ! **(I)** We obtain the results of existing approaches using the released open source code. **(II)**  The methods we compared (like Tune-A-Video, FateZero and RAVE) are all based on the **same CLIP text encoder** that enables us to integrate the same input concept pairs into the models. For MotionDirector, it supports a **spatial path** to bring the object of concept video into latent space **originally**. That is to say, we can make fair comparisons for existing approaches. **(III)** Besides, we release the code of merging representation of object in concept video into the CLIP text encoder in our anonymous repository https://anonymous.4open.science/w/STIVE-PAGE-B4D4/
>
> ```separate quantitative comparison table with and without concept video```
> Thanks for the suggestion. We seperate the quantitative comparison into two part for using and not using concept videos to make it more clear as shown bellow. We will include this in the revision latter.
>
> For editing w/ concept video
>
> |  | M-PSNR | Concept Consistency | Frame Consistency |
> | --- | ---- | ---- | ---- |
> | **Tune-A-Video** | 14.70 | 0.6982 | 0.9399 |
> | **FateZero** | 17.08 | 0.6822 | 0.9413 |
> | **MotionDirector** | 12.73 | 0.7222 | 0.9452 |
> | **RAVE** | 17.39 | 0.6990 | 0.9379 |
> | **Ours** | **19.71** | **0.7642** | **0.9472** |
>
> For editing w/o concept video
>
> |  | M-PSNR | Frame Consistency |
> | --- | ---- | ---- |
> | **Tune-A-Video** | 15.72 | 0.9397 |
> | **FateZero** | 19.42 | 0.9246 |
> | **MotionDirector** | 16.86 | 0.9403 |
> | **RAVE** | 16.20 | 0.9306 |
> | **Ours** | **22.10** | **0.9405** |
>
> ```Missing references and some for comparisons```
>
> Thanks. These methods each have unique characteristics, primarily focusing on innovations in pretrained models, diffusion models, attention mechanisms, spatiotemporal consistency, and inter-frame relationships in video editing. For example,  [1] **AnyV2V** uses a pretrained model to handle various video editing tasks, while [2] **GenVideo** employs a target-image-aware approach and novel InvEdit masks to overcome text-prompt limitations. Besides, [3] **Factorized Diffusion Distillation** introduces the EVE model by distilling pretrained diffusion models. For temporal consistency, [4] **VidToMe** merges self-attention tokens across frames. For spatial editing, [5] **Pix2Video**, [6] **Slicedit**, and [8] **Cideo-P2P** both improve results with attention features and diffusion models respectively. [7] **TokenFlow** leverages inter-frame correspondences to propagate features. Compared to existing approaches, our method focuses on **attention supervision and control mechanisms** and operates on a **one-shot video editing** paradigm. It also improves **temporal consistency** through extended temporal module parameters and enables the flexible integration of **external concept objects**. This approach not only differs from existing methods in task handling strategies but also complements them in principles and implementation details, thereby broadening the possibilities for video editing.
>  We will include these references and make comparisons in the revision.
>
> ```Detailed explanation for the pipeline of editing without concept video```
>
> Thanks. A pretrained lora/concept model is not required for editing without concept video. For example, we only use the word embedding in CLIP model to edit the "jeep" for "Porsche", then fine-tuning the denosing net with SCAM loss for source video and source prompt, and TCAM loss for source video and target prompt.
>
> ```The range of concepts in CATI for generalization```
>
> Good question ! Our experiment shows that CATI can be applied to various objects, like robots and animals. In fact, it's not that difficult for objects in concept image or video to **generalize well** to new image or video. But for **stable video editing** (high consistency of non-target area and high matching faithfullness between source and target object), it is more suitable to edit **rigid objects**, like vehicles or other **objects with less inner component movement**.

---

### Official Review · Reviewer_yeHN · 2024-11-06

**Soundness:** 2
**Presentation:** 2
**Contribution:** 2
**Rating:** 3
**Confidence:** 4

**Summary:**

This paper proposes a text-driven video editing method using Stable Diffusion. It involves concept-augmented textual inversion and a dual prior supervision mechanism. The results show that the proposed method can outperform the baselines.

**Strengths:**

1. The overall presentation of the paper is easy to follow. Most of the details are clear and well-documented.

2. Based on the quantitative metrics, the model can outperform the previous baselines.

**Weaknesses:**

1. My main concern is that the overall quality of the edited videos is not satisfactory. Mainly, there are obvious changes (shape and color) between frames, so the videos don’t look consistent. I think this may be the inherent weakness of the approach of inflating Stable Diffusion with temporal layers. The base model doesn’t have video prior knowledge compared to a video diffusion model which has been trained on video datasets. Possible ways to improve can be adding motion prior such as adopting optical flow [1] or propagating features among frames [2].

2. Besides the quality, the methodology is also not strong to me. Textual inversion comes with several variations, and one of them can be finetuning the model parameters instead of only finetuning the textual embedding. It seems concept-augmented textual inversion belongs to this variation by additionally finetuning V. The scam and tcam loss, although improves the results, does not seem novel or strong enough to boost the consistency of the videos.

3. I appreciate that the authors also provide details of efficiency. I think the efficiency is also not optimal enough. More than 30 minutes (training + inference) to edit a 6-frame video seems inefficient and not practical.

[1] Cong, Yuren, et al. "Flatten: optical flow-guided attention for consistent text-to-video editing." ICLR 2024.

[2] Geyer, Michal, et al. "Tokenflow: Consistent diffusion features for consistent video editing." ICLR 2024.

**Questions:**

1. What are the parameters that are finetuned during the scam and tcam loss optimization?

2. What possible reasons do you think that the edited videos are not looking consistent enough? What possible solutions could you propose?

---

> ### Author Response · Authors · 2024-11-23
>
> We thank Reviewer yeHN  for the constructive reviews. We address your concerns bellow.
>
> ```overall quality of the edited videos is not satisfactory.```
>
> Good question. We acknowledge that the edited video does indeed contain some of the aspects you have carefully observed. But we guess the reason behind this lies in that we set a larger frame sampling step due to limited computational resources (**only one RTX 4090 with 24G gpu memory**) instead of only lacking video prior knowledge. Because, **(I)** we have actually validated video diffusion model at the very beginning. However, while the results demonstrate marginal advantages in the temporal, they fell far short of our expectation for scene editing (especially for spatial control). Therefore, we opt to use the standard Stable Diffusion coupled with temporal layers after careful consideration. **(II)** We conduct additional experiments with a small frame step with more powerfull devices (Nvidia L20 with 48G memory) in **scenarios with weak temporal consistency**, the edited videos are more smoother than before as shown in **More Results Section** of https://anonymous.4open.science/w/STIVE-PAGE-B4D4/. While our original experiments used a frame sampling step of 8 and sequence length of 6 frames, we test our approach with a streamlined step of 3 frames (**i**) and an extended sequence length of 14 frames (**ii**). The comparative analysis reveals the following:
>
> for editing w/ concept video
> |  | CLIP-L | DINOv2 ViT-G | LPIPS VGG |
> |  -  | ----  | ------  |  ----  |
> | i | 0.9309 | 0.8234 | 0.6795 |
> | ii | **0.9475** | **0.8966** | **0.7026** |
>
> for editing w/o concept video
> | | CLIP-L | DINOv2 ViT-G | LPIPS VGG |
> |  -  | ----  | ------ | ---- |
> | i | 0.9169 | 0.8217 | 0.6793 |
> | ii | **0.9410** | **0.8579** | **0.7075** |
>
> **(III)** Although Flatten [1] guides video motion editing by introducing optical flow conditions, during inference, Flatten retains prior information such as convolutional and attention features from certain diffusion steps, which are then replaced during the denoising process to guide video editing. However, this process conflicts with operations such as attention swapping, **making Flatten unsuitable for  this method**. TokenFlow [2] involves propagating estimated motion tokens to modify hidden states after extended attention computation, is fundamentally incompatible with our approach. Specifically, this propagation mechanism conflicts with our attention control strategy during the inference phase, **making TokenFlow framework unsuitable for integration into our method**.
>
> ```The methodology is not strong for (I) concept-augmented textual inversion belongs to this variation by additionally finetuning V; (II) the scam and tcam loss although improves the results, does not seem novel or strong enough to boost the consistency of the videos.```
>
> Thanks for the very detailed comments. **(I)** We acknowledge that our approach is quite straightforward. However, it is not merely about replacing model parameters with V. More importantly, our analysis reveal that fine-tuning the denoising network in one-shot video editing scenarios is highly prone to **overfitting** and incurs **significant computational costs**. In contrast, fine-tuning V effectively addresses these issues, offering a simple yet efficient solution tailored to this specific task. We believe this is novel. **(II)** We propose the scam and tcam loss to **improve the consistency** in non-target areas before and after video editing, especially in editing source video with concept video, which is illustrated in Tab.1 and Fig.3 in the draft. This is also appreciated by the reviewer ZGWC "SCAM and TCAM is novel".
>
> ```The efficiency is not optimal enough.```
>
> Good question ! But we would like to clarify that **(I)** It seems unfair to challenge efficiency because for the same setting : Tune-A-Video needs 10 min with GPU A100; MotionDirector needs 20 min with GPU A6000 (48G); while **our approach needs about ~30 min with only one RTX 4090 (24G)** **(II)** Exisiting approach like MotionDirector achieves a lower resolution for 384x384, rather than 512x512 in other methods. **(III)** We further validate our approach with Nvidia L20 (48G) which shows that our approach spends about ~20 min, which is on par with existing methods but achieves more compelling results (Tab.1 & Fig.3).

---

> > ### Comment · Reviewer_yeHN · 2024-11-27
> >
> > Thank the authors for the detailed reply and the extra experiments. I also appreciate that the authors have tried video model backbones and also analyzed the two papers I mentioned.
> >
> > While acknowledging that the presented quality in the paper is limited by the computational resources, I still tend to believe that the overall methodology is lack of enough novelty, and the performance is not strong in terms of visual quality or computational efficiency. In other words, there is not enough contribution that would make me accept this paper. The authors could try to analyze more about how and why finetuning V is efficient than simply tuning the whole model or the other parts of the models, which would make the contribution claim stronger.

---

> ### Author Response · Authors · 2024-11-24
>
> Dear Reviewer yeHN,
>
> Hoping this message finds you well. Your comments and the provided insights are very valuable to our work. We will attach these analysis and additional experiments to our final version. As the rebuttal-discussion period is nearing its end, could you please review our response (especially in https://anonymous.4open.science/w/STIVE-PAGE-B4D4/) to see if it addresses your concerns? Your timely feedback will be extremely valuable to us. Could you read and let us know if there are more questions? We would be very grateful! Your decision is of utmost importance to us, and we earnestly hope that you will consider the contribution of this research to the field of one shot video editing. Thank you very much!
>
> Best regards,
>
> The Authors

---

> ### Author Response · Authors · 2024-11-28
>
> We thank the reviewer for their feedback and for recognizing the effort we put into additional experiments and analysis. We appreciate the acknowledgment of our exploration of video model backbones and the analysis of the mentioned papers.
>
> **(1) On Novelty and Contribution:**
> While we respect your perspective on the novelty of the methodology, we would like to reiterate that our approach addresses a **unique and underexplored challenge in concept-driven one-shot video editing**. Specifically:
>
> (a) The **selective fine-tuning of V projection weights in the attention mechanism** is a targeted and efficient solution to integrate novel concepts while preserving consistency and stability.
>
> (b) The use of **concept pairs (text and video)** introduces a new level of controllability, enabling precise embedding of specific target objects from a reference video into a source video, which purely text-based methods cannot achieve.
>
> (c) Our method strikes a practical balance between computational efficiency and visual quality, a crucial consideration for real-world applications.
>
> **(2) Visual Quality and Computational Efficiency:**
> We acknowledge the reviewer's concerns about performance in terms of visual quality and computational efficiency. The limitations in visual quality primarily stem from the inherent challenges of one-shot video editing and the need to handle diverse and arbitrary concepts. Despite this, our results show significant improvement over existing baselines, and we believe this demonstrates the effectiveness of our approach.
>
> **(3) Fine-tuning V Projection Weights:**
> We appreciate the suggestion to delve deeper into why fine-tuning the V weights is efficient compared to tuning the entire model or other parts. The key insight lies in:
>
> **(a) Stability and Focused Adaptation:** Fine-tuning V allows the model to integrate new feature representations into the pre-trained attention mechanism while minimizing disruptions to the pre-existing distribution, thereby enabling stable training.
>
> **Efficiency:** By restricting updates to V, we reduce the number of trainable parameters, achieving a lower VRAM overhead without sacrificing performance.
>
> **Task-Specific Design:** This selective tuning is particularly suited to our one-shot video editing task, where preserving the pre-trained model's knowledge while adapting to new concepts is crucial.
>
> We will incorporate a more detailed analysis of these aspects in future revisions to strengthen our contribution claims. Thank you again for your valuable feedback. We deeply appreciate your constructive suggestions and will use them to further improve our work.

---

### Official Review · Reviewer_oUvi · 2024-11-06

**Soundness:** 2
**Presentation:** 2
**Contribution:** 2
**Rating:** 5
**Confidence:** 3

**Summary:**

This paper introduces an improved text-driven video editing approach using generative diffusion models. It overcomes limitations of traditional methods by incorporating a concept-augmented textual inversion technique and a dual prior supervision mechanism. For the concept-augmented textual inversion, LoRA is adopted, and for the dual prior supervision mechanism, a cross-attention-based masking technique is utilized.

**Strengths:**

* Quantitative/qualitative evaluation shows relatively improved performance.
* The two proposed methods intuitively make sense.

**Weaknesses:**

* Writing needs improvement. It is difficult to distinguish what is being proposed and what are existing components in the method section.

* Lack of technical novelty. I think the use of LoRA with Textual Inversion (or other inversion/personalization technique) is already widely used as open-source, and compared to these, the use of LoRA in the proposed Concept-Augmented Textual Inversion does not appear to be significantly different. Also, although it is minor, it is not convincing why this technique is called Concept-Augmented Textual Inversion.

* In Dual Prior Supervision, a mask is obtained by thresholding the activated attention part corresponding to the text via cross-attention, and this is applied to the loss. This approach itself has been widely used in segmentation and image editing (e.g., masactrl) and does not appear technically novel. Even setting aside the issue of technical novelty, I have concerns about the approach of specifying areas corresponding to certain words through hard thresholding, which seems heuristic and may have a marginal effect outside situations where the object is explicitly referenced.

**Questions:**

Please see the weaknesses.

---

> ### Author Response · Authors · 2024-11-23
>
> We thank Reviewer-oUvi for the comments. **Reviewer-oUvi thinks "The two proposed methods intuitively make sense" but gives a reject rating due to questioning the core methodology ?**  We address  Reviewer-oUvi's concerns bellow.
>
> ```Reviewer-oUvi criticized our writing "needs improvement" -- for example, not clearly  stating about the core proposed components.```
>
> Thanks. We kindly **disagree** with the comment. We already **highlight** our motivation and propose **Concept-Augmented Textual Inversion** and **Dual Prior Supervision** at Line 61-69. **(I)** For **Concept-Augmented Textual Inversion**,  we enable one-shot flexible video editing based on external word embedding and target video (what we term concept prompt and concept video), we further advance the textual inversion tailored for concept-augmented video editing by addressing the under-fitting issue inside the original approach. (See lines 199-235)  **(II)** We propose **Dual Prior Supervision** (SCAM and TCAM) to supervise the attention between word embedding and unrelated areas in video to maintain consistency. (See lines 236-273). **(III)** Both reviewers appreciate our draft "**clear writing**", "**easy to follow**". _**May we humbly ask for more specific details that support 'writing needs improvement'?**_
>
>
> ```Novelty of our Concept-Augmented Textual Inversion (CATI)```
>
> Thanks. We would like to emphasize that the reviewer **may have confused the distinction between our motivation and the specific implementation**. We aim to flexibly edit source videos using concept videos as a key task setting, with our primary motivation being to enrich the latent space through one-shot concept videos to achieve more faithful inversion representations. We validate this improvement through ablation studies, as shown in Fig. 4. We deeply appreciate the reviewer's feedback and would like to clarify that **the proposed CATI method is technically novel**. While LoRA and Textual Inversion have been widely used **separately** in image style transfer tasks, to the best of our knowledge, no existing variant of textual inversion simultaneously employs LoRA tailored for video editing task. Existing approaches that do fine-tune the denoising network face significant challenges in one-shot concept video scenarios, such as overfitting and high memory requirements during training. Our approach addresses these limitations by leveraging LoRA modules to fine-tune attention value weights with a smaller learning rate, maintaining **low VRAM overhead** during tuning while preserving **plug-and-play** capabilities. _**As also appreciated by Reviewer-HLfd "The motivation is clear". May we kindly request citations if similar motivations exist?**_
>
>
> ```Novelty of Dual Prior Supervision (mask generation by thresholding the activated attention, specifying areas corresponding to certain words through hard thresholding )```
>
> Thanks. **(I)** We would like to clarify that **there appears to be a misunderstanding regarding the mask generation process**. The mask is actually obtained through a pretrained OWL-ViT model, not by thresholding the activated attention, as detailed in Lines 252-254 of our paper. We appreciate the mention of MasaCtrl, which is indeed an excellent work for consistent image synthesis and editing using mutual self-attention control. While previous works like Prompt-to-Prompt and FateZero have made valuable contributions with cross-attention control and self-attention blending control methods, our method serves as a supervision mechanism designed to **reduce inconsistencies in non-target areas** before and after video editing, addressing the low consistency in existing approaches, especially in editing source video with concept video (As illustrated in Tab.1 in our paper. **The reviewer ZGWC also mentioned that our SCAM and TCAM is novel**). **(II)** Noted that Dual Prior Supervision **does not specify** areas corresponding to certain words through hard thresholding. **Instead**, the SCAM and TCAM losses are designed to optimize only the attention probability between unrelated areas and certain words toward zero, without imposing constraints on the probability between related areas and the corresponding words. This mechanism is formally described in Eq.(5) and Eq.(6) of our paper.

---

> > ### Comment · Reviewer_oUvi · 2024-11-23
> >
> > Thank the authors for providing detailed responses. I will carefully review the replies and the manuscript again and get back as soon as possible.

---

> ### Comment · Reviewer_oUvi · 2024-11-24
>
> Thank you for your detailed response and for pointing out some parts I had missed. Regarding the Dual Prior Supervision, the reviewer's response has convinced me and alleviated my concerns about this part.
>
> First, I would like to clarify my understanding to ensure we are on the same page: CATI utilizes textual inversion, a technique commonly employed in T2I models, to internalize the concept of a user-provided video for video editing. In doing so, instead of merely applying textual inversion, the authors incorporated LoRA to address the issue of underfitting. (Please correct me if my understanding is wrong.)
>
> My concerns are: (1) Using LoRA to avoid underfitting is a method commonly employed in T2I models, and thus does not offer new technical contributions when adapting to video editing (https://github.com/cloneofsimo/lora); (2) Employing textual inversion to internalize a concept video is a straightforward approach, and it may be challenging to consider this innovative. There might not be substantial technical insights to gain from this aspect.
>
> I do acknowledge the authors' conceptual contribution of utilizing concept videos, irrespective of the implementations. However, as reviewers yeHN, G4M5, ZGWC, and I have similarly pointed out, there is concern that this method may lack sufficient technical contribution. The notable difference in the authors' proposed LoRA+textual inversion tailored for video editing appears to be the selective training of the V projection LoRA. It would be beneficial to emphasize these aspects in the paper.
> Overall, CATI leans more towards a conceptual contribution rather than specific architectural design choices. However, I feel that the current manuscript could improve in clearly articulating these contributions. For readers not deeply engaged with this field, the introduction may seem somewhat vague. For instance, terms like "concept template" (Line 71) and phrases such as "constrained by the size of the latent space" (Line 59) required me to do multiple readings to comprehend. Does the "latent space" mentioned refer to the space of text embedding features? I recommend that the authors include a figure prior to the current Figure 1 to better highlight the conceptual novelty distinguishing this work from existing approaches.  It would be also beneficial to include detailed explanations of the newly added inputs: concept video and concept prompt and how they relate to existing inputs: the source video, source prompt, and target prompt, throughout the introduction and method sections.
>
> Given that my concerns regarding the Dual Prior Supervision section have been addressed, I am revising my score to a borderline reject.

---

> ### Author Response · Authors · 2024-11-24
>
> Thank you very much for your thoughtful and constructive feedback. We greatly appreciate your patience and the time you've taken to carefully review our work. We are pleased to hear that our clarifications regarding the Dual Prior Supervision have alleviated your concerns, and we will address the points you raised below.
>
> ```About lack sufficient technical contribution.```
>
> We deeply value the reviewer's feedback, but we believe CATI we proposed has technical contribution.
> Compared to our method, the conventional Textual Inversion approach **lacks novel object information in arbitrary concept videos** which causes word embeddings obtained through inversion having **insufficient fidelity** in describing target objects. Directly porting Textual Inversion to one-shot video editing scenario struggles from (always failed) generating videos for novel provided concept pairs. Our approach addresses these limitations by utilizing cutting-edge LoRA modules to fine-tune attention value weights (We latter explain why we only tune V), effectively maintaining the advantages of **low VRAM overhead** during tuning while preserving **play-and-plug** capabilities.
>
> As for **V projection LoRA for training**, our goal is to perform inversion while integrating **novel object information in arbitrary concept videos** for one-shot video editing task. Textual inversion process relies on the pre-trained denoising-net established text-image attention probability distribution to achieve accurate target representation. In this context, fine-tuning only the V weights instead of Q and K allows new feature representations to be directly integrated while **suppressing changes towards the pre-trained attention distribution** to enable stable training during the early stages.
>
>  ```"latent space"```
>
> Thanks. In our work, the term "**latent space**" does not refer to the text embedding feature space. Instead, it refers to the **latent space of the latent diffusion model** in [1], specifically the spatial-level lower-dimensional latent representation produced by the variational autoencoder during the generative process. This latent space is a compact representation of the input data and serves as the working space for the diffusion model's operations.
> When we mention "**constrained by the size of the latent space**" we are referring to the fact that the **capacity** of the diffusion model's latent space is **inherently limited by the scale and diversity of the image samples used during pre-training**. This limitation (see explanations for previous question) affects the representation power of the latent space, consequently, the quality of the inversion results. Relying solely on Textual Inversion to obtain a word embedding from concept an image or video often leads to **insufficient representation capabilities**, especially for samples far from dataset for pre-training.
>
> [1] Rombach R, Blattmann A, Lorenz D, et al. High-resolution image synthesis with latent diffusion models[C]//Proceedings of the IEEE/CVF conference on computer vision and pattern recognition. 2022: 10684-10695.
>
> ```"concept template" & Details of how concept video and concept prompt relate to existing inputs.```
>
> Good questions! The template here means we could change the source video object with several different concept videos coupled with specified objects to enable more diverse and flexible one shot video editing. For example,
>
> - source prompt : "a **man** rides a wooden skateboard on the handrail of the staircase with arms outstretched"
> - concept prompt - I : "a **OPTIMUS** is squatting on a wooden floor with its back to a white wall"
> - target prompt - I :  "a **OPTIMUS** rides a wooden skateboard on the handrail of the staircase with arms outstretched"
> - concept prompt - II:   "**NEO** is outstretching his arms"
> - target prompt - II :  "a **NEO** rides a wooden skateboard on the handrail of the staircase with arms outstretched"
> - concept template:  "a 【**CONCEPT**】 rides a wooden skateboard on the handrail of the staircase with arms outstretched"
>
> For current source video and prompt, when employing different target pairs we can obtain diverse editing results based on concept tamplate. For more details, please refer to  Fig.3(setting I) and Fig.13.

---

### Meta-Review · Area_Chair_XBFv · 2024-12-16

**Metareview:**

This paper receives reviews of 5,3,6,3,3. The AC follows the decisions of the reviewers to reject the paper. The main concerns of this paper are: 1) The need to improve on the writing to make the technical contributions clearer. 2) The lack of novelty since the use of LoRA with Textual Inversion is already widely used as open-source. 3)  The overall quality of the edited videos is not satisfactory. There are obvious inconsistencies between the frames of the videos. 4) Evaluation metrics and comparison methods are unconventional and somewhat outdated. On the other hand, the reviewer who gave a 6 did not give any strong reasons to overrule the majority of the other reviewers who reject the paper.

**Additional Comments On Reviewer Discussion:**

Although the authors tried to clarify some of the weaknesses, the reviewers remain unconvinced and did not raise their scores. The AC thus follow the majority decision to reject the paper.

---

### Decision · Program_Chairs · 2025-01-22

Reject